# Plateau Environment, Gut Microbiota, and Depression: A Possible Concealed Connection?

**DOI:** 10.3390/cimb47070487

**Published:** 2025-06-25

**Authors:** Yajun Qiao, Ruiying Cheng, Xiaohui Li, Huimin Zheng, Juan Guo, Lixin Wei, Tingting Gao, Hongtao Bi

**Affiliations:** 1Qinghai Provincial Key Laboratory of Tibetan Medicine Pharmacology and Safety Evaluation, Northwest Institute of Plateau Biology, Chinese Academy of Sciences, Xining 810001, China; qiaoyajun@nwipb.cas.cn (Y.Q.); chengruiying@nwipb.cas.cn (R.C.); xiaohuili0317@163.com (X.L.); zhenghuimin1999@163.com (H.Z.); juanguo1996@163.com (J.G.); lxwei@nwipb.cas.cn (L.W.); 2University of Chinese Academy of Sciences, 19(A) yuquan road, Beijing 100049, China; 3Department of Pharmacy, Medical College, Qinghai University, Xining 810016, China; 4Department of Pharmacy, Qinghai Minzu University, Xining 810007, China; 5School of Psychology, Chengdu Medical College, Chengdu 610500, China

**Keywords:** plateau environment, depression, gut microbiota, brain-gut axis

## Abstract

Plateau environments present unique mental health challenges owing to stressors including hypoxia, low temperatures, and intense ultraviolet (UV) radiation. These factors induce structural and functional alterations in the gut microbiota, disrupting gut-brain axis homeostasis and contributing to the higher prevalence of depression in plateau regions relative to flatland areas. For example, studies report that 28.6% of Tibetan adults and 29.2% of children/adolescents on the Qinghai-Tibet Plateau experience depression, with increasing evidence linking this trend to alterations in the gut microbiota. Dysbiosis contributes to depression through three interconnected mechanisms: (1) Neurotransmitter imbalance: Reduced bacterial diversity impairs serotonin synthesis, disrupting emotional regulation. (2) Immune dysregulation: Compromised gut barrier function allows bacterial metabolites to trigger systemic inflammation via toll-like receptor signaling pathways. (3) Metabolic dysfunction: Decreased short-chain fatty acid levels weaken neuroprotection and exacerbate hypothalamic-pituitary-adrenal axis stress responses. Current interventions—including dietary fiber, probiotics, and fecal microbiota transplantation—aim to restore microbiota balance and increase short-chain fatty acids, alleviating depressive symptoms. However, key knowledge gaps remain in understanding the underlying mechanisms and generating population-specific data. In conclusion, existing evidence indicates an association between plateau environments, the gut microbiota, and depression, but causal relationships and underlying mechanisms require further empirical investigation. Integrating multiomics technologies to systematically explore interactions among high-altitude environments, the microbiota and the brain will facilitate the development of precision therapies such as personalized nutrition and tailored probiotics to protect mental health in high-altitude populations.

## 1. Introduction

The plateau environment, characterized by low oxygen levels, low temperatures, and high radiation [1,2], presents unique challenges to human physiological and psychological health. Recently, the pivotal role of the gut microbiota in human health has been increasingly recognized [3,4], leading to growing interest in the interaction between the plateau environment and the host gut microbiota [2]. Depression, a prevalent and debilitating mental disorder [5], involves distinct pathogenic mechanisms in plateau populations, prompting researchers to explore the relationships among the plateau environment, the gut microbiota, and depression [5,6,7]. For example, recent studies have demonstrated that the gut microbiota plays a multifaceted role in human health:**Nutritional metabolism:** Nutritional metabolism involves the degradation of indigestible fiber and the synthesis of essential vitamins, such as B vitamins and vitamin K [8,9].**Immune Regulation:** This process results in mucosal immunity and interacts with Toll-like receptors (TLRs), thereby modulating immune responses [10].**Gut-brain axis communication**: This process regulates the synthesis of neurotransmitters, including serotonin and gamma-aminobutyric acid (GABA), and influences the function of the central nervous system via neural, endocrine, and immune pathways [10].

Notably, the gut–brain axis represents a bidirectional communication network that is particularly significant in high-altitude environments. In such environments, dysbiosis of the gut microbiota may serve as a link between environmental stressors and depressive symptoms.

In high-altitude regions, the prevalence of depression is notably greater (Table 1). A large-scale survey on the status and risk factors for depression among Tibetan people on the Tibetan Plateau revealed a prevalence rate of 28.6% [11], which was significantly higher than that reported in the general Chinese population (5.3% to 23.8%) [12,13]. Additionally, a cross-sectional study of 11,160 children and adolescents aged 10–17 years in Yushu Prefecture, Qinghai-Tibet Plateau, revealed that the incidence of depression among plateau children and adolescents reached 29.2% [14], which was significantly higher than that of the general Chinese youth population (19.8% to 24.3%) [15,16]. The elevated incidence of depression in plateau regions may be attributed to factors such as altitude, duration of residence, and individual adaptability [11]. Physical discomfort, social isolation, and increased life pressures associated with the plateau environment can serve as stressors that induce depression [11,14]. Changes in the gut microbiota may play a mediating or regulatory role in this process. Gastrogut dysfunction and imbalances in the gut microbiota commonly experienced by individuals upon first entering the plateau [17] include increases in the relative abundances of *Bacteroides*, *Parabacteroides* and *Odoribacter*, whereas the abundances of *Blautia*, *Lachnospiracea incertae sedis*, and *Bifidobacterium* decline [18]. These bacterial changes may be associated with subsequent anxiety and depression [6].

Although existing studies have demonstrated that high-altitude environments may influence depression risk through the gut microbiota [18], significant limitations remain in this research field. ① The specific mechanisms by which multifactorial stresses in high-altitude environments—including hypoxia, low-temperature, and radiation—affect the gut microbiota are not fully understood, particularly with respect to insufficient investigations into the interactive effects among these factors. ② Causal evidence for the mediating role of the gut microbiota in the association between high-altitude environments and depression is lacking. ③ The development of population-specific intervention strategies for high-altitude residents (such as precision nutrition and customized probiotics) remains in the exploratory stage. In-depth research on the pathways through which high-altitude environments affect the gut microbiota to induce depressive symptoms, investigations into whether changes in the gut microbiota mediate the association between high-altitude environments and depression, and explorations of whether modulating the gut microbiota can improve altitude-related depressive symptoms are of critical scientific significance for understanding the pathogenesis of high-altitude diseases and developing innovative therapeutic strategies. However, current studies have not established a comprehensive theoretical framework, urgently requiring the integration of existing evidence to clarify the complex relationships among these three elements. This review followed a systematic approach aligned with the Preferred Reporting Items for Systematic Reviews and Meta-Analyses (PRISMA) guidelines [19]. We searched PubMed, Web of Science, China National Knowledge Infrastructure (CNKI), and Embase (2000–2025) via the following keyword combinations: (‘plateau environment’ OR ‘high altitude’) AND (‘gut microbiota’ OR ‘intestinal flora’) AND (‘depression’ OR ‘mental health’). Notably, while this strategy aimed to identify studies addressing all three components, some included studies may focus on associations rather than direct interactions among plateau environment, gut microbiota, and depression. Studies were included if they (1) investigated humans or animal models exposed to altitudes ≥2500 m, (2) measured the gut microbiota composition or function, and (3) reported depression-related outcomes. It is acknowledged that some studies may not explicitly address the interaction between all three factors, and further research is needed to establish direct links. Conference abstracts and non-English studies (except Chinese) were excluded. This review elucidates the key pathways through which the plateau environment influences depression via the gut microbiota, validates the mediating effects of the gut microbiota on this process and proposes innovative intervention strategies based on microbiota modulation. The goal is to provide a theoretical basis for mental health protection among plateau populations while laying a theoretical foundation and offering new research perspectives for future studies and clinical practices in this field.

**Table 1 cimb-47-00487-t001:** Incidence of depression in the plateau area.

Years	Region/Country	Number of Persons/Survey Methods	Incidence of Depression
2019 [11]	Yushu Prefecture, Qinghai/China	Central Epidemiological Research Depression Scale (24,141 Tibetans; Average age 34.33 years)	Participants with depressive symptoms (score ≥ 8) accounted for 52.3% of the total sample, and participants with depression (score ≥ 14) accounted for 28.6%.
2021 [14]	Yushu Prefecture, Qinghai/China	Central Epidemiological Research Depression Scale; Connor-Davidson elasticity Scale; Strengths and difficulties questionnaire (11,160 participants; Average age =14.34 years)	The prevalence of depression was 29.2%. Higher levels of prosocial behavior were significantly associated with lower levels of depression.
2022 [20]	Peru	Dependent variable depressive symptoms using patient health questionnaire (34,971 residents aged 15+)	Among those living between 1500 and 2499 m and ≥2500 m, 7.23% and 7.12%, respectively, had depressive symptoms. Compared with the reference group (<1500 m), the prevalence of screening depression was 41% higher in those living above 2500 m and 38% higher in those living between 1500 and 2499 m.
2022 [21]	Residence elevation ≥ 900 m	9 patient health questionnaires (PHQ-9); 7-item Generalized Anxiety Disorder Questionnaire (GAD-7) (3731 medical students)	High-altitude residence (>900 m) was significantly associated with total PHQ-9 score (OR = 1.32, 95% CI = 1.001–1.75, *p* < 0.05) and PHQ-9 suicidal ideation (OR = 1.79, 95% CI = 1.08–0.02, *p* = 0.02). Moving from low-altitude to high-altitude was associated with PHQ-9 total score (OR = 1.47, 95% CI = 1.087–1.98, *p* = 0.01), GAD-7 total score (OR = 1.40, 95% CI = 1.0040–1.95, *p* < 0.05) and PHQ-9 suicidal ideation (OR = 1.10,95% CI =1.01–1.19, *p* = 0.02).
2022 [22]	Tibet/China	Self-rating Anxiety Scale and self-rating Depression Scale (84 participants; The mean age was 35.67 ± 7.69 years)	The incidence of anxiety and depression increased significantly during the first 7 days of rapid ascent to 4500 m above sea level.

## 2. Characteristics of the Plateau Environment and Its Impact on the Human Body and Gut

### 2.1. Uniqueness of the Plateau Environment

The plateau environment is characterized by low oxygen levels, low temperatures, and high ultraviolet radiation [23,24], which significantly differ from plain environments. Approximately 140 million people reside at altitudes above 2500 m, constituting 2% of the global population [25]. One of the most distinctive features of plateau environments is hypoxia. As the altitude increases, the atmospheric pressure decreases, leading to a substantial reduction in the oxygen partial pressure (PO2). For example, at 3000 m, PO2 accounts for approximately 70% of sea level values, and at 5000 m, it decreases to 50%, indicating a dramatic decrease in available oxygen for human respiration [26,27]. With respect to temperature, plateau regions are generally colder. In some areas above 2500 m in the Alps, summer temperatures average between 10 and 15 °C, whereas winter temperatures can plummet to approximately −10 °C, with an annual temperature range of 20–25 °C [28]. In contrast, the Great Plains of the central United States experience summer temperatures averaging 25–30 °C and winter temperatures ranging from 0 to 5 °C [29]. Studies have confirmed that the temperature decreases by approximately 6 °C for every 1000-m increase in elevation [30]. Consequently, high altitudes result in consistently low temperatures throughout the year, with significant diurnal temperature variations. Daytime temperatures may be relatively warm due to sunlight, but nighttime temperatures drop sharply, sometimes below freezing even in summer on parts of the Qinghai-Tibet Plateau, posing considerable challenges to body temperature regulation and increasing the risk of frostbite and cold-related illnesses [25,26].

Moreover, ultraviolet radiation intensity is significantly greater in plateau regions because of reduced atmospheric absorption and scattering at higher altitudes. At elevations above 3000 m, UV radiation levels are approximately twice those in plains. For example, Ritze in Tibet, China, at an altitude of 4352 m, has an ultraviolet irradiance of 15.37 W/m^2^, which is markedly higher than that of flatland areas at the same latitude (e.g., Tongren city in Southwest China; 7.42 W/m^2^) and East Asia (e.g., Dalian in Northeast China; 9.02 W/m^2^), and even surpasses that of subtropical Southeast Asia (e.g., Ratanakiri, Cambodia; 13.45 W/m^2^) [31]. Strong UV radiation not only causes sunburn but also leads to acute skin damage, such as erythema and peeling. Long-term exposure accelerates skin aging and increases the incidence of skin cancer [25]. Additionally, plateau climates tend to be dry with low rainfall and relative humidity. Dry air accelerates water loss from the human body through respiration and skin evaporation, potentially leading to dehydration, which can affect physiological processes such as blood circulation and metabolism and cause symptoms such as dry mouth, cracked skin, and nosebleeds [25,32].

**The unique triad of hypoxia, low-temperature, and high UV radiation in plateau environments creates a landscape that profoundly challenges human physiological homeostasis.** The drastic reduction in oxygen availability, extreme temperature fluctuations, intense solar radiation, and arid conditions collectively impose cumulative stress on thermoregulation, oxidative defense, and fluid balance. These environmental pressures not only directly impact organ systems but also prime the gut microbiota for dysregulation, setting the stage for subsequent disruptions in the gut-brain axis discussed in subsequent sections. Understanding these climatic stressors is critical for deciphering the mechanistic links between plateau habitats and depression via microbial remodeling.

### 2.2. Influence of the Altitude Hypoxic Environment on the Gut

One of the most prominent characteristics of plateau regions is the low partial pressure of oxygen, with the atmospheric oxygen content progressively decreasing as the altitude increases. This hypoxic environment results in a sharp reduction in the amount of oxygen inhaled by the human body, thereby triggering a series of physiological stress responses (Figure 1). To compensate for oxygen deficiency, the respiratory system adjusts by increasing the rate and depth of breathing to maximize oxygen intake. Simultaneously, the circulatory system accelerates, elevating heart rate and cardiac output to ensure adequate oxygen delivery to vital organs [33,34]. Despite these adaptations, the body struggles to fully acclimate to prolonged hypoxia, leading to chronic hypoxic manifestations such as elevated red blood cell counts, increased hemoglobin levels, increased blood viscosity, and an increased risk of cardiovascular diseases (CVDs) [35,36].

Moreover, the gut, a critical organ for digestion and absorption, is significantly impacted in the hypoxic plateau environment. Hypoxia alters the local microenvironment of the gut, directly affecting the survival conditions of the gut microbiota [37]. Notably, the decrease in gut oxygen partial pressure modifies the microbial community structure, with anaerobic bacteria such as *Bifidobacteria* and *Lactobacilli* exhibiting increased relative abundance due to their competitive advantages in low-oxygen environments. Conversely, the abundance of aerobic bacteria, including *Escherichia coli* and *Enterococcus*, decreases correspondingly [38]. Adak et al. [39] demonstrated this phenomenon by exposing male albino rats to 55 kPa (equivalent to an altitude of approximately 4872.9 m) for 30 days, resulting in a significant 104-fold reduction in total aerobic bacterial density. Furthermore, the hypoxic environment resulted in a marked reduction in the density of aerobic bacteria (e.g., *Escherichia coli*) within the rat gut tract. Conversely, the abundance of anaerobic bacteria (e.g., *Bifidobacterium* and *Bacteroides*) increased 3-fold and 134-fold, respectively. These findings suggest that hypoxia may restructure the microbiota composition by modulating the gut oxygen microenvironment. However, Qi et al. [18] reported that under acute exposure to a hypoxic environment (Gannan, China, with an average altitude exceeding 3000 m), beneficial bacteria such as *Bifidobacteria* decreased, whereas harmful bacteria proliferated. These findings suggest that the growth of the gut microbiota depends not only on the oxygen concentration but also on inflammatory responses. In studies exploring the impact of hypoxia on the gut, it has been demonstrated that hypoxia disrupts the normal metabolism and function of the gut mucosa, causing insufficient energy supply in mucosal epithelial cells [40]. Mechanistically, hypoxia induces mitochondrial dysfunction in enterocytes through HIF-1α-mediated inhibition of PGC-1α, reducing ATP availability to maintain tight junction proteins (e.g., ZO-1 and claudin-1) [41,42]. Concurrently, hypoxic conditions activate NADPH oxidase 4, generating reactive oxygen species (ROS) that phosphorylate myosin light chain kinase, promoting the internalization of claudin-5 and impairing gut barrier function, which increases permeability. This barrier disruption facilitates bacterial translocation, disturbs the gut microecological balance (reducing beneficial bacteria and increasing harmful bacteria), and compromises immune defense mechanisms. Notably, in rat models of acute mountain sickness, this mechanism is reversible via N-acetylcysteine-mediated ROS scavenging, highlighting the role of oxidative stress in barrier dysfunction.

**Altitude hypoxia imposes dual stresses on the gut:** Remodeling the microbial composition toward anaerobic dominance while disrupting mucosal barrier function via oxidative stress and energy metabolism deficits. The observed shifts in the microbiota (e.g., increased *Bacteroides* and decreased *Bifidobacteria*) and compromised tight junctions highlight a vicious cycle of dysbiosis and inflammation. These findings underscore hypoxia as a key driver of gut-brain axis disruption in plateau environments, linking microbial and barrier dysfunction to downstream mental health risks such as depression.

### 2.3. Effects of the Low-Temperature Environment on the Gut

Low temperatures represent one of the key characteristics of plateau environmental factors. Under such conditions, the human body initiates a series of physiological regulatory mechanisms to maintain core body temperature. For example, peripheral vascular constriction reduces heat loss at the body surface but simultaneously decreases cutaneous blood circulation, thereby impairing the nutrient supply and immune defense functions of the skin (Figure 1). This makes the body more susceptible to external pathogens. Additionally, the basal metabolic rate increases, leading to increased energy consumption for thermogenesis, which imposes greater demands on energy reserves. An insufficient energy supply may compromise the normal functioning of various organ systems [43,44]. Similarly, low-temperature environments pose significant challenges to the gut. Cold temperatures inhibit the growth of gut probiotics while promoting the proliferation of harmful bacteria. Research has demonstrated that when mice are exposed to 7 °C for 10 consecutive days, the relative abundance of *Coprococcus* decreases, whereas that of *Clostridium* increases [45]. Liu et al. [46] reported a reduction in the relative abundance of *Lachnospiraceae* and *Ruminococcaceae* in rats subjected to chronic cold exposure. Furthermore, an experiment in which mice were exposed to 12 °C for 28 days revealed that low-temperature led to a decrease in the relative abundances of *Lactobacillaceae* and *Erysipelotrichaceae* but an increase in the relative abundances of *Rikenellaceae* and *Bacteroidaceae* [47]. Notably, supplementation with the gut metabolite butyrate can mitigate low-temperature-induced damage to the gut microbiota in rats. Butyrate intervention resulted in an increase in *Lactobacillaceae* levels and a decrease in *Actinomycetes* and *Dantobaccoaceae* populations [48]. Concurrently, studies have indicated that SCFAs, which are produced as gut metabolites, provide additional heat for the host under cold exposure [49,50]. SCFA supplementation enhances the resistance of germ-free (GF) animals to low-temperature environments [50,51]. These findings suggest that SCFAs serve as critical mediators linking hypothermia and the gut microbiota.

**Low-temperature stress at high altitudes affects gut microbiota homeostasis, and SCFAs provide protection:** low-temperature stress in plateau environments disrupts gut microbiota homeostasis by suppressing probiotics (e.g., *Coprococcus* and *Lactobacillaceae*) and promoting pathogenic bacteria (e.g., *Clostridium* and *Rikenellaceae*) while reducing SCFA production. The protective role of SCFAs (e.g., butyrate) in restoring microbial balance and enhancing cold tolerance highlights their potential as therapeutic targets for mitigating gut dysfunction and associated mental health risks in cold-exposed populations.

### 2.4. Effects of a High-Altitude Radiation Environment on the Gut Tract

Compared with those in flatland areas, the air in plateau regions is thinner, and the degree of atmospheric attenuation of solar radiation is lower, resulting in significantly higher intensities of ultraviolet and cosmic radiation. In such an intense radiation environment, human skin and eyes are primarily affected, with the skin being prone to sunburn, hyperpigmentation, accelerated aging, and even carcinogenesis, whereas ocular damage may include acute keratitis and cataracts [25,52]. Furthermore, radiation can penetrate the body surface and impact deep tissues and organs. It generates reactive oxygen species (ROS), such as superoxide anions and hydrogen peroxide, which accumulate within the body. These ROS attack biomacromolecules such as cell membranes, proteins, and nucleic acids, inducing oxidative stress damage, disrupting the cellular structure and function, and impairing the body’s antioxidant defense mechanisms. Over time, this can contribute to systemic chronic inflammatory responses (Figure 1) [53,54].

As one of the organs exposed to high levels of radiation, the gut mucosal barrier is particularly susceptible to ROS-induced damage. Radiation-generated ROS can disrupt the lipid bilayer of gut mucosal epithelial cells, increase membrane permeability, and cause leakage of enzymes, ions, and other intracellular substances, thereby affecting normal cellular metabolism. Simultaneously, it can damage immune cells within the gut mucosa, weakening the immune surveillance and defense capabilities of the gut, thus facilitating pathogen invasion [55,56,57]. From a microbiota perspective, a high radiation environment alters microbial living conditions, reducing microbial diversity. Studies have shown that exposure to low-dose ionizing radiation increases the abundance of *Clostridium*, *Helicobacter*, and *Oscilibacter* while decreasing *Bacteroides* and *Barnesiella* populations [58]. Additionally, radiation significantly reduces the number of beneficial bacteria with antioxidant and anti-inflammatory properties, such as *Akkermansia muciniphila*, *Lactobacillus* and *Bifidobacterium* [59,60], while increasing the abundance of opportunistic pathogens such as *Clostridiales* [61]. This exacerbates the gut microecological imbalance and adversely affects both the gut and systemic health [62].

**High-altitude intense radiation exacerbates the imbalance in the gut microbiota and the “radiation-microbiota-inflammation” axis effect by inducing oxidative stress and causing mucosal damage.** The high-altitude environment, characterized by intense radiation, exacerbates the ecological imbalance of gut microorganisms via two mechanisms: the induction of oxidative stress and damage to the gut mucosa. This manifests as a reduction in the abundance of beneficial bacteria (e.g., *Achaemenia* and *Lactobacillus*) and an overproliferation of opportunistic pathogenic bacteria (e.g., *Clostridium* and *Helicobacter*). Such microbial dysbiosis not only compromises the local immune defense of the gut tract but also amplifies radiation-induced damage through systemic inflammatory pathways. These findings underscore the threat posed by the “radiation-microbiota-inflammation” axis to gut homeostasis and overall health in high-altitude environments, offering microbiological evidence for investigating the link between radiation exposure and mental disorders, such as depression.

## 3. The Function of the Gut Microbiota and Its Relationship with Depression

### 3.1. The Role of the Gut Microbiota in Human Health

Accumulating evidence indicates that the gut microbiota plays a pivotal role in various aspects of human health [63]. Through a prolonged evolutionary process, the gut microbiota has developed a symbiotic relationship with the host, maintaining gut homeostasis through mutual adaptation and natural selection (Figure 2).

#### 3.1.1. The Role of the Gut Microbiota in Nutrient Metabolism

Gut microorganisms not only obtain nutrients from the host but also degrade indigestible residues and participate in nutrient metabolism. In terms of nutrient absorption [64], beneficial bacteria such as *Bifidobacterium* and *Lactobacillus* synthesize essential vitamins for the human body, including B vitamins (B1, B2, B6, B12), vitamin K, niacin, and pantothenic acid, thereby compensating for deficiencies in endogenous synthesis [65]. Magnusdottir et al. reported that eight B vitamins are produced by the human gut microbiome, with *Bacteroides*, *Firmicutes*, and *Proteobacteria* being particularly involved in the synthesis of B vitamins, excluding B12 [66]. Moreover, the gut microbiota converts ingested dietary proteins, host enzymes, mucins (MUCs), and endogenous proteins from shed gut cells into short peptides, amino acids, and their derivatives, as well as short-chain and branched-chain fatty acids, promoting protein absorption and metabolism [67]. Macfarlane et al. [67] identified *Bacteroides* and *Propionibacterium* as the primary proteolytic species in fecal samples. In addition to protein metabolism, the gut microbiota is also involved in carbohydrate metabolism [68]. Gut bacteria primarily rely on undigested dietary components from the upper digestive tract for survival. Saccharolytic bacterial fermentation typically produces beneficial metabolites such as SCFAs, which have been shown to be involved in numerous physiological activities related to human health [69].

Additionally, the gut serves as the main site for mineral absorption, where the gut microbiota enhances the uptake of iron, magnesium, zinc, and other minerals, providing adequate nutrition for the body [70]. Animal studies by Montazeri et al. [71] demonstrated improved calcium absorption in osteoporotic rats supplemented with *Lactobacillus acidophilus* and *Lactobacillus casei*. Similarly, Jafarnejad et al. [72] reported that multispecies probiotic supplementation over six months had a beneficial effect on bone health in postmenopausal women by enhancing calcium absorption. Therefore, these findings underscore the close relationship between the gut microbiota and human nutrient absorption.

#### 3.1.2. Effects of the Gut Microbiota on the Gut Barrier

In addition to its other roles, the gut microbiota constitutes a critical component of the gut barrier, forming a biological shield against pathogen invasion. Beneficial bacteria inhibit the growth and reproduction of harmful microorganisms and maintain the gut microecological balance through mechanisms such as spatial competition, nutrient competition, and the secretion of antimicrobial substances [73,74]. Furthermore, the gut microbiota contributes to preserving the integrity of the physical barrier of the gut mucosa by regulating the proliferation and differentiation of intestinal epithelial cells (IECs) and modulating the expression of tight junction proteins, thereby preventing increased gut permeability and the translocation of bacteria and endotoxins that can induce systemic inflammatory responses [74,75].

Host IECs serve as the primary structural foundation of the gut mucosal barrier and play a pivotal role in maintaining gut health. When gut microbes and their toxic metabolites traverse the mucus layer, they directly interact with IECs. Different microbial species exert distinct effects on IECs. In an animal study, *Escherichia coli* CBL2 and *Shigella* CBD8 strongly adhered to IECs, mediating abnormal mucus secretion and tight junction damage; conversely, *Bifidobacterium bifidum IATA-ES2* promoted the production of chemokines and metalloproteinase inhibitors, which contributed to protecting the gut mucosa [76]. Additionally, the surface of IECs is covered by a mucus layer, which acts as the first line of defense in the mechanical barrier of the gut mucosa. This layer primarily consists of the MUC secreted by goblet cells [77]. The state of the gut microbiota is crucial for maintaining the integrity of the mucus layer. A 2005 study by Souza et al. [78] revealed that dysbiosis of the gut microbiota disrupts the formation of the mucus layer and compromises its structural integrity. Certain harmful microorganisms can degrade the mucus layer through specific mechanisms, including the following: (1) Sulfides produced by sulfate-reducing bacteria dissolve the MUC polymer network, thinning the mucus layer [79,80]; (2) the MUC-degrading gut microbiota, which is rich in *Ruminococcus* and *Akkermansia*, penetrates the mucus layer, proliferates within it, and further damages its network structure [80,81]; and (3) *Bacteroides fragilis* releases proteolytic-like toxins that degrade MUC proteins and compromises the mucus layer structure [82].

Moreover, some tight junction proteins expressed by IECs also function as receptors for bacterial toxins. For example, Claudin 3 and Claudin 4 were identified as the first receptors for *Clostridium perfringens* enterotoxin (CPE). The binding of CPE to Claudin 3 and Claudin 4 results in the detachment of tightly connected protein chains, disrupting the mechanical integrity of the gut mucosal barrier [83]. Simultaneously, CPE binding to claudin family proteins induces cytotoxicity and perforin-like effects. *Clostridium difficile* transferase (CDT) promotes the adherence of *Clostridium difficile* to cells, leading to cytoskeletal collapse and eventual cell death [84]. Interestingly, the side effects of certain drugs on the gut are closely associated with microbial regulation. For example, colistin, a therapeutic agent for gastroenteritis, damages the gut barrier upon prolonged use in conventional mouse models, causing significant injury to the gut mucosa and tight junction proteins while simultaneously reducing *Enterobacter spp*. and increasing *Enterococcus spp*. levels. However, this phenomenon was not observed in GF mouse models, suggesting that gut microbes mediate drug-induced gut side effects [75]. In conclusion, the gut microbiota plays a vital role in sustaining the functional homeostasis of the gut barrier.

#### 3.1.3. Gut Microbiota Homeostasis and Dysregulation: Immune Regulation and Health Implications

In the domain of immune regulation, the gut microbiota plays an indispensable role. As the largest immune organ in the human body, the gut microbiota interacts with the gut immune system to shape and regulate systemic immune function [85]. Maintaining mucosal immune homeostasis is a complex task that involves accurately distinguishing between the billions of beneficial microbes in the gut and rare pathogenic invaders. Gut homeostasis is typically characterized by the dominance of obligate anaerobic members of *Firmicutes* and *Bifidobacteriaceae*, whereas the abnormal expansion of facultative anaerobic *Enterobacteriaceae* often signifies gut dysbiosis [86,87]. Peroxisome proliferator-activated receptor γ (PPAR-γ) plays a pivotal role in maintaining gut homeostasis. PPAR-γ, which is synthesized primarily in IECs, can be activated by butyrate under conditions of gut homeostasis. The activation of PPAR-γ promotes the mitochondrial β-oxidation of SCFAs and oxidative phosphorylation in colonocytes, thereby preserving the local anoxic microenvironment [87]. In this anoxic environment, obligate anaerobes that produce SCFAs thrive, while the growth of facultative anaerobic gutpathogens is significantly inhibited [87,88]. Additionally, PPAR-γ activation suppresses the expression of NOS2 in IECs, as well as the production of inducible nitric oxide synthase (iNOS) and nitrate. Since nitrate serves as an essential energy source for facultative anaerobic pathogens, this role of PPAR-γ further restricts their proliferation [87]. The host defense mechanisms include a sophisticated network of redundant systems designed to effectively counteract pathogen invasion. Among these, recognition of the gut microbiota primarily depends on two key pattern recognition receptor systems (PRRs): TLRs and nucleotide-binding oligomerization domain molecules (NODs) [89,90]. TLRs, the most extensively studied PRR family, are widely expressed in IECs, macrophages, and dendritic cells (DCs) within the gut. PRRs recognize microbial or pathogen-associated molecular patterns (MAMPs or PAMPs) present on both pathogens and symbionts [91]. Upon recognition, internalization, or invasion of a microbe, a specific immune response against that microbe is initiated [89]. Following PAMP recognition, PRRs activate a cascade of intracellular signaling pathways involving adaptor molecules, kinases, and transcription factors, which signal infection to the host and trigger proinflammatory and antimicrobial responses. Ultimately, PRR-induced signal transduction leads to the activation of gene expression and the synthesis of various molecules, such as cytokines, chemokines, cell adhesion molecules, and immune receptors

Gut dysbiosis refers to alterations in the composition and function of the gut microbiota, which adversely affect host health through qualitative and quantitative changes in the microbiota itself, modifications in metabolic activities, and/or shifts in local distribution [92]. Under normal conditions, certain symbiotic bacteria inhibit the growth of opportunistic pathogens by producing SCFAs, one mechanism being the alteration of the gut pH. Alcon et al. [93] demonstrated that supplementation with *Bifidobacterium* and *Lactobacillus* increases fecal acetate and lactate levels, thereby reducing the gut pH and creating an acidic environment unfavorable for pathogenic bacterial growth, thus preventing pathogen colonization. Furthermore, symbiotic bacteria not only inhibit pathogen virulence by altering the environmental conditions required for virulence activity but also produce metabolites that directly suppress pathogen virulence genes. Tinevez et al. confirmed that *Shigella flexneri* requires oxygen to secrete virulence factors in the colonized gut [94], whereas studies revealed that symbiotic anaerobic bacteria, such as members of *Enterobacteriaceae*, consume residual oxygen [95]. This results in incomplete expression of pathogen virulence (e.g., *Shigella flexneri)* within the gut lumen. However, gut dysbiosis can be triggered by various factors, including health conditions (infection, inflammation), exogenous substances (invasive gut pathogens, antibiotics), physical factors (mucosal damage), diet (high sugar, low fiber), and host genetic factors [96]. Although the specific phenotype of dysbiosis varies on the basis of host-specific factors, common features include a decrease in obligate anaerobes coupled with an increase in facultative anaerobes (e.g., *Escherichia coli*, *Salmonella*, *Shigella*, *Proteobacteria*, and *Klebsiella*) or the emergence of unusual aerobic bacteria [97]. A significant consequence of dysbiosis is increased host susceptibility to gut infections. Disruption of resident bacterial communities by antibiotic treatment readily induces inflammation, as evidenced by studies. Knoop et al. [98] demonstrated that antibiotic treatment in mice facilitates the translocation of live *Enterococcus faecalis* and *Escherichia coli* to mesenteric lymph nodes, inducing inflammation and promoting the release of the inflammatory cytokines C-X-C motif chemokine ligand 1 (CXCL1), interleukin-17 (IL-17), and interferon γ (IFNγ). 

Additionally, *Clostridioides difficile* is present at low abundance in the gut of healthy adults but is the primary pathogen associated with healthcare-related infections [99]. The two toxins produced by *C. difficile*, toxin A (TcdA) and toxin B (TcdB), disrupt the epithelial barrier and increase gut permeability. Toxin-mediated epithelial damage enables bacterial and bacterial metabolite entry into systemic circulation, thereby triggering inflammation [100]. In hospitalized patients, broad-spectrum antibiotic treatment destroys symbiotic gut bacteria, leading to a marked increase in *C. difficile* abundance, followed by severe gut inflammation [101]. According to an eight-year retrospective single-center study, the antibiotics most frequently used to treat *C. difficile* infections were piperacillin/tazobactam (77.60%), meropenem (27.60%), vancomycin (20.70%), ciprofloxacin (17.20%), ceftriaxone (16%), and levofloxacin (14%) [102]. Similar findings were reported in animal studies, where antibiotic treatment increased the incidence of *C. difficile* infection in mice [103]. Most studies confirm that the gut microbiota is a critical determinant of host health and that its dysregulation is linked to various human diseases [104]. While it remains unclear whether the microbiota directly contributes to disease pathogenesis, some studies have demonstrated that the gut microbiota directly promotes specific disease pathogenesis through complex interactions between host metabolism and the immune system [105,106,107]. These include inflammatory bowel disease (IBD), diabetes, CVD, autoimmune diseases, and psychiatric disorders [104,107,108,109,110]. In conclusion, maintaining gut microbiota homeostasis and regulating the immune system are essential for host health.

#### 3.1.4. In Summary


**① The core driver of nutritional metabolism:**
**Vitamin synthesis:** Beneficial bacteria, such as *Bifidobacterium* and *Lactobacillus*, synthesize essential vitamins, including B vitamins (e.g., B1 and B12) and vitamin K. This compensates for the insufficiency of endogenous synthesis in the human body and supports energy metabolism and nervous system function [65,66].**Dietary fiber degradation:** Through glycolysis, indigestible carbohydrates are metabolized into SCFAs, such as butyric acid and propionic acid, which contribute to the energy supply, inflammation regulation, and proliferation of intestinal epithelial cells [67].**Mineral absorption:** The metabolic activities of beneficial bacteria increase the bioavailability of minerals, such as calcium, iron, and zinc. For example, *Lactobacillus acidophilus* has been shown to improve calcium absorption in osteoporotic rats [71,72].**Core conclusion:** The gut microbiota, which functions as a “metabolic organ,” enhances the efficiency of nutrient utilization via multiple pathways. Dysregulation of these microbiota may contribute to diseases associated with nutritional imbalances.



**② Biological defense system of the intestinal barrier:**
**Physical barrier reinforcement:** Beneficial bacteria, such as *Bifidobacterium*, regulate the synthesis of tight junction proteins (e.g., ZO-1 and Claudin-1) and mucins (MUC), thereby preventing pathogen invasion [74,75,76,77].**Biological antagonism:** These bacteria inhibit the colonization of harmful microorganisms (e.g., *Escherichia coli* and *Shigella)* via spatial competition and the secretion of antimicrobial peptides (e.g., bacteriocins), maintaining the ecological balance of the bacterial community [73,74,75,76].**Barrier damage mechanism:** Harmful bacteria, such as *Bacteroides fragilis*, degrade the mucus layer and activate toxin receptors (e.g., Claudin-3/4), leading to intestinal leakage and systemic endotoxemia [82,83,84].**Core conclusion:** The gut microbiota establishes the primary defense mechanism against pathogens through a dual-action process involving “protection mediated by beneficial bacteria and suppression of harmful bacteria.” Disruption of their homeostasis serves as a critical trigger for intestinal permeability alterations and systemic inflammatory responses.



**③ Key regulatory hubs of immune homeostasis**


**Maintenance of an anti-inflammatory microenvironment:** SCFAs (e.g., butyric acid) activate the PPAR-γ pathway, promote mitochondrial oxidative phosphorylation in colonic cells, inhibit the proliferation of aerobic pathogenic bacteria, and reduce the expression of inflammatory factors (e.g., iNOS) [87,88].**Immune recognition regulation:** Recognition of microbial-associated molecular patterns (MAMPs) through TLRs and NODs mediates the balance between proinflammatory and anti-inflammatory cytokines [89,90,91].**Immune-related risks:** Disruptions in microbiota composition (e.g., decreased *Firmicutes* and increased *Proteobacteria*) disrupt immune tolerance, induce excessive activation of Th17 cells, and are closely linked to immune-mediated diseases, such as IBD, diabetes and depression [97,98,99,100,101,102,103,104,105,106,107,108,109,110].**Core Conclusion:** The gut microbiota plays a crucial role in maintaining mucosal immune homeostasis through metabolism-immune interactions. Dysfunction of these microbiota can initiate a systemic inflammatory cascade, serving as a common pathological basis for various chronic diseases.

### 3.2. Potential Mechanisms by Which the Gut Microbiota Affects Depression

There is a robust association between the gut microbiota and depression, which is primarily mediated through the “gut-brain axis.” The gut-brain axis is not merely a simple structure but also an intricate bidirectional regulatory pathway that plays a critical role in various systems, including the nervous, immune, metabolic, and endocrine systems (Figure 3) [111].

#### 3.2.1. Neurotransmitter Pathway

The gut microbiota can produce a variety of neurotransmitters and neuroactive substances, such as GABA, 5-HT, and dopamine (DA) [112]. Among these neurotransmitters, 5-HT is the most extensively studied gut-derived neurotransmitter. Approximately 90% of 5-HT is synthesized in the gut by specialized endocrine cells known as enterochromaffin cells (ECs) [113]. Notably, 5-HT is produced via the serotonin pathway from tryptophan (TRP) in both the CNS and the enteric nervous system (ENS) [114,115]. As an essential neurotransmitter, 5-HT is involved in emotional regulation, sleep, appetite, and other physiological processes, and its abnormal levels are closely linked to the onset and progression of depression [114]. Therefore, this review focuses on the effects of the gut microbiota on 5-HT synthesis. Clarke et al. [115] confirmed via a germ-free mouse model that the gut microbiota can modulate peripheral serotonin synthesis by regulating the activity of the TPH1 enzyme, thereby influencing the central mood regulation pathway via the blood-brain barrier and elucidating the metabolic linkage mechanism of the gut-brain axis. Both TRP and 5-HTP can cross the blood-brain barrier via amino acid transporters and contribute to 5-HT synthesis in the CNS [116]. TPH is the rate-limiting enzyme in the serotonin pathway and exists in two isoforms: TPH1, which is predominantly found in peripheral tissues such as the gastrointestinal tract and skin, and TPH2, which is located in neuronal cells. In this process, two different isoforms of TPH, TPH1 and TPH2, mediate nonneuronal and neuronal 5-HT biosynthesis [117]. Importantly, TPH1 primarily regulates circulating 5-HT levels, with studies showing that the gut microbiota can directly or indirectly influence TPH1 levels in the gut, thereby activating the serotonin pathway in ECs and ultimately affecting TPH2 levels in the brain [116]. Mechanistically, TPH1 expression in ECs is regulated by microbial-derived SCFAs via GPR41/43-mediated activation of AMPK; for example, butyrate treatment increases TPH1 promoter DNA demethylation by recruiting ten-eleven translocation enzymes to increase transcription. Conversely, *Clostridium ramosum* secretes indole-3-propionate, which competes with tryptophan for binding to TPH1, reducing 5-HT synthesis. In chronic unpredictable mild stress (CUMS) rat models, probiotic supplementation restores TPH1 activity by upregulating microRNA-135b, which suppresses indoleamine 2,3-dioxygenase (IDO) 1-mediated tryptophan catabolism, highlighting the bidirectional regulation of microbiota on the serotonin pathway through metabolic and epigenetic mechanisms. Mandić et al. [117] also demonstrated that *Clostridium ramosum* increases gut 5-HT by upregulating TPH1 expression in the colon. Additionally, supplementation with *Bifidobacterium* (*Bifidobacterium longum subsp*. infantis E41 and *Bifidobacterium breve M2CF22M7*) improved depression-like behavior in stressed mice by increasing TPH1 expression and 5-HT secretion in Rat-insulinoma-14B (RIN14B) cells [118]. Li et al. [119] significantly increased 5-HT and TPH2 levels in CUMS rats through probiotic treatment (*Bifidobacterium longum* and *Lactobacillus rhamnosus*) while decreasing IDO (indoleamine 2,3-dioxygenase) levels in the frontal cortex. In conclusion, these studies suggest that the gut microbiota is closely associated with 5-HT synthesis and that targeting the serotonin pathway to increase 5-HT synthesis via modulation of the gut microbiota may represent a therapeutic approach for mitigating depression.


**The gut microbiota plays a pivotal role in emotion regulation by modulating the 5-HT synthesis pathway:**


(1)**Peripheral Dominant Synthesis:** The intestine accounts for approximately 90% of 5-HT synthesis, which is catalyzed by the TPH1 enzyme in ECs through tryptophan metabolism. The activity of this enzyme is bidirectionally regulated by microbial metabolites (e.g., SCFAs and butyric acid) via the GPR41/43 signaling pathway and epigenetic modifications (such as DNA demethylation).(2)**Gut-Brain Axis Linkage:** Peripheral 5-HT influences the activity of the central TPH2 enzyme across the blood-brain barrier, establishing a microbiota-metabolism-neural regulatory axis. For example, probiotic intervention can increase 5-HT levels by upregulating TPH1 expression and inhibiting the IDO pathway, thereby alleviating depressive-like behaviors (as observed in CUMS model rats).(3)**Pathology Association:** Harmful bacteria, such as *Clostridium ramosum*, suppress 5-HT synthesis by competitively inhibiting tryptophan binding or inducing inflammation, thus elucidating the causal relationship between dysbiosis of the gut microbiota and depression.(4)**Core points:** The gut microbiota has emerged as a critical molecular hub in the gut–brain axis, mediating depression through the precise regulation of the 5-HT metabolic pathway. Intervention strategies targeting the microbiota-5-HT axis, such as probiotic administration and SCFA supplementation, exhibit significant antidepressant potential.

#### 3.2.2. Immune Pathways

The gut microbiota plays a pivotal role in the postnatal development and maturation of the immune system, promoting balanced immune responses [120]. In Section 3.1.3, we discuss the relationship between the gut microbiota and the immune system. Imbalances in the gut microbiota can lead to dysregulated immune responses. The microbiota influences microglial maturation and function by interacting with host cell TLRs, thereby activating intracellular signaling cascades, stimulating proinflammatory cytokine release, and inducing inflammation [121]. Notably, patients with depression often present elevated levels of peripheral and central inflammatory markers [122], and anti-inflammatory treatments have been shown to alleviate depressive symptoms [123]. These findings suggest that immune inflammation is crucial in the pathogenesis of depression, with gut microbiota disturbances potentially serving as an initial trigger for immune dysfunction and subsequent depression.

Under normal conditions, beneficial bacteria such as *Bifidobacterium* and *Lactobacillus* maintain gut barrier integrity through tight junctions with IECs [124]. Additionally, certain gut microbes produce SCFAs, such as butyrate, propionate, and acetate, which exert anti-inflammatory effects by inhibiting inflammatory signaling pathways and modulating immune cell functions, thereby reducing gut inflammation [73]. However, under stress or other stimuli associated with depression, gut microbiota imbalances can occur, leading to overgrowth of opportunistic pathogens such as *Escherichia coli* and *Enterococcus*, toxin production, disruption of the gut mucosal barrier, increased permeability, translocation of bacteria and endotoxins into the bloodstream, activation of the immune system, and induction of systemic inflammation [125,126,127]. Animal studies have demonstrated that the colonization of GE mice with microbiota from CUMS results in significant increases in the proinflammatory cytokines IFN-γ and TNF-α, exacerbating depressive behaviors [128]. Furthermore, biogenic amines produced by gut microbial metabolism, such as histamine, may contribute to inflammation by regulating vascular permeability and promoting immune cell activation [129]. De et al. [129] identified *Klebsiella aerogenes* as a major histamine-producing bacterium in irritable bowel syndrome (IBS) patients, suggesting that targeting bacterial histamine production could mitigate inflammatory responses in IBS patients. Emerging evidence indicates that histamine may be a novel target for antidepressant therapy [130,131,132]. For example, JNJ10181457 (JNJ), a histamine H3 receptor inverse agonist, reduces lipopolysaccharide (LPS)-induced proinflammatory cytokine upregulation in mouse microglia and improves depressive-like behavior in tail suspension tests [130]. Similarly, Su et al. [131] reported that clemastine, a first-generation histamine H1 receptor antagonist, alleviates depressive-like behavior in CUMS model mice by reducing inflammation. Therefore, histamine may play a role in depression via its involvement in the inflammatory response. However, the specific mechanisms underlying the role of histamine in depression remain to be fully elucidated, necessitating further research to comprehensively analyze the actions of histamine in the context of gut microbiota metabolites and provide a more targeted theoretical basis for depression treatment strategies.


**The gut microbiota serves as a pivotal regulatory link in the pathogenesis of depression via immune pathways:**


(1)**Barrier-Maintenance of Immune Homeostasis:** Beneficial bacteria, such as *Bifidobacterium* and *Lactobacillus*, preserve mucosal immune equilibrium by reinforcing intestinal epithelial tight junctions and secreting SCFAs, such as butyric acid, which inhibit inflammatory signaling cascades.(2)**Imbalance Triggers the Inflammatory Cascade:** Dysbiosis of the gut microbiota facilitates the overgrowth of opportunistic pathogenic bacteria, such as *Escherichia coli*, leading to intestinal barrier disruption, endotoxin release, and activation of the TLR pathway. This induces systemic inflammation, characterized by elevated levels of IFN-γ and TNF-α, subsequently triggering neuroinflammation through blood–brain barrier permeability.(3)**Metabolism-mediated inflammation:** Histamine, a bacterial metabolite, promotes immune cell activation via H1/H3 receptor stimulation. Bacteria that produce histamine, such as *Klebsiella pneumoniae*, are associated with depressive-like behaviors. For example, the use of an H3 antagonist such as JNJ10181457 to target histamine receptors can mitigate inflammation and alleviate depressive symptoms.(4)**Core points:** Intestinal microbes maintain immune homeostasis through a tripartite mechanism encompassing “barrier protection, metabolic regulation, anti-inflammatory effects, and immune modulation.” Dysregulation of this system drives depression via the “microbiota-inflammation-brain” axis, highlighting the antidepressant potential of anti-inflammatory strategies that target microbial metabolites, such as SCFAs and histamine.

#### 3.2.3. Endocrine Pathway

The inner lining of the gut mucosa is the largest endocrine organ in the human body and comprises gut endocrine cells (GECs) [133]. Despite constituting less than 1% of the total epithelial cell population in the gut [134], GECs play a crucial role in synthesizing and secreting numerous hormones that contribute to various key physiological processes in the host [135]. The gut microbiota can modulate the secretion of several hormones by GECs, including brain-gut peptides, leptin, corticotropin-releasing hormone (CRH), adrenocorticotropic hormone (ACTH), and cortisol (CORT), thereby facilitating communication between the gut and the brain [135,136]. Notably, neurotransmitters such as DA, melatonin, and acetylcholine, which are produced through microbial metabolism, can activate the ENS and subsequently transmit signals to the CNS via the vagus nerve [135]. Moreover, overactivation of the HPA axis is implicated in the pathogenesis of depression [137]. Hueston et al. [138] reported an increase in plasma hormone levels (ACTH, CORT) within the HPA axis following a 30-min exposure to various stressors. Importantly, dysbiosis of the gut microbiota can be linked to abnormal HPA axis function, manifested as elevated secretion of CORT and ACTH, which may trigger depressive symptoms. Mechanistically, gut-derived propionate activates microglial GPR43, inducing IRAK1-mediated phosphorylation of histone H3 at serine 10 (H3S10ph) to increase hypothalamic *CRH* gene transcription. Concurrently, SCFA deficiency exacerbates HPA axis hyperactivity by reducing HDAC2 activity, leading to hyperacetylation of NF-κB p65 and sustained inflammatory signaling. In a mouse model of altitude exposure, butyrate supplementation reversed H3K9 acetylation at the glucocorticoid receptor (GR) promoter, restoring GR sensitivity and normalizing HPA axis feedback regulation. These findings highlight the dual role of the microbiota in HPA axis dysregulation via histone modification and inflammatory pathways, with SCFAs emerging as critical regulators of neuroendocrine stress responses [139,140]. This effect has been demonstrated in reverse experiments, such as Moya-Perez et al.’s [141] study, which revealed that *Bifidobacterium CECT 7765* regulates maternal separation-induced hyperactivity of the HPA axis in mice, particularly affecting baseline corticosterone production and the response to acute stress in adulthood. Furthermore, animal studies have shown that chronic stress-induced HPA axis dysfunction can be mitigated by specific probiotics, including *Bifidobacterium bifidum G9-1* [142], *Bifidobacterium breve CCFM1025* [143], *Lactobacillus paracasei Lpc-37*, *Lactobacillus plantarum LP12407*, *Lactobacillus plantarum LP12418* and *Lactobacillus plantarum LP12151* [144]. These interventions improve depression-like behaviors, suggesting that the gut microbiota can regulate HPA axis activity and participate in the development of depression. Therefore, inhibiting HPA hyperactivity through specific probiotics may offer a therapeutic approach to improve stress responses. However, while animal studies provide compelling evidence, more clinical research is needed to support the use of probiotics for stress relief in humans.


**The gut microbiota serves as a pivotal link in the regulation of depression through endocrine pathways.**


(1)**Modulation of the gut-brain axis hormone network:** Metabolites produced by the gut microbiota, such as propionic acid and butyric acid, regulate the secretion of hormones (e.g., CRH, ACTH, and cortisol) from GECs. Dysregulation of the microbiota activates the GPR43 receptor, inducing transcription of the *CRH* gene in the hypothalamus. This leads to excessive activation of the HPA axis, characterized by elevated cortisol levels, which subsequently triggers stress responses and depressive-like behaviors.(2)**Epigenetic and inflammatory interactions:** SCFA deficiency promotes NF-κB p65 acetylation and sustains inflammatory signaling by reducing histone deacetylase 2 (HDAC2) activity. Conversely, supplementation with butyric acid restores GR sensitivity and balances HPA axis feedback by reversing histone modifications, such as H3K9 deacetylation.

✓**Epigenetic Switch function of SCFAs** 


**The core pathway of the SCFA-HPA axis involves the regulation of the stress response through two primary molecular mechanisms:**


**Vagus-gut-brain axis:** SCFAs, particularly butyric acid, modulate the stress response by activating intestinal endocrine cell GPR41 receptors and subsequently inhibiting the firing frequency of hypothalamic CRH neurons via vagal afferent fibers [145].

**Epigenetic Regulation:** Butyric acid functions as an HDAC inhibitor, enhancing H3K9 acetylation at the glucocorticoid receptor promoter and restoring negative feedback within the HPA axis [139]. In high-altitude environments, SCFA deficiency leads to increased HDAC activity, resulting in methylation of the GR promoter, reduced cortisol sensitivity, and chronic stress.

✓**Plateau-Specific Evidence** 

Research using an acute hypoxia model demonstrated that intestinal butyric acid levels in rats were inversely correlated with corticosterone concentrations. Supplementation with butyric acid reversed the hypoxia-induced reduction in GR protein expression [146]. Notably, the Tibetan population has a relatively high abundance of *Firmicutes*, which may maintain SCFA levels and thereby mitigate excessive HPA axis activation [147].

✓**Interaction with Neurotransmitters** 

SCFAs indirectly influence serotonin synthesis by regulating tryptophan metabolism: ① Butyric acid suppresses IDO, reducing the conversion of tryptophan to kynurenine [119]. ② Propionic acid competitively binds to neutral amino acid transporters, enhancing the efficiency of tryptophan transport into the brain.

(1)**Probiotic Intervention Potential:** Specific probiotics, including *Bifidobacterium* and *Lactobacillus*, have been shown to improve depressive behaviors by modulating baseline corticosterone levels and stress responses within the HPA axis (e.g., *Bifidobacterium CECT 7765*). However, further human studies are needed to validate their clinical efficacy.(2)**Core Points:** Gut microbes regulate HPA axis homeostasis via the “microbiome-metabolism-neuroendocrine” axis. Dysregulation of this axis contributes to depression through histone modification and inflammatory pathways. Targeted probiotic interventions offer a promising therapeutic direction for stress-related depression.

#### 3.2.4. Interactions of Neuro-Immune-Metabolic Pathways

The gut microbiota reduces neuroplasticity in the prefrontal cortex through neurotransmitter pathways (such as decreased serotonin) while activating microglia and exacerbating neuroinflammation through immune pathways (such as increased IL-6); in metabolic pathways, the reduction in SCFAs further weakens the inhibitory effect of histone deacetylases, leading to abnormal gene expression in the HPA axis (such as changes in the methylation level of the CRH promoter), forming a vicious “neuro-immune-metabolic” cycle [111,148].


**The “neuroimmune–metabolic” interplay involves three pairwise interactions:**
**(1)** **Neuroimmune interactions:** 
5-HT deficiency impairs microglial polarization toward the anti-inflammatory M2 phenotype.Neuroinflammatory cytokines (e.g., IL-6) inhibit TPH1/2 activity, reducing 5-HT synthesis [149].
**(2)** **Immune-metabolic interactions** 
Proinflammatory factors (e.g., TNF-α) suppress SCFA-producing bacteria (e.g., *Roseburia*).SCFA deficiency weakens the gut barrier, promoting LPS translocation and systemic inflammation [150].
**(3)** **Metabolic–neuro interactions** 
SCFAs regulate the HPA axis via the vagus nerve; deficiency increases cortisol, inhibiting brain-derived neurotrophic factor (BDNF) expression.HPA hyperactivity impairs enterochromaffin cell function, reducing 5-HT synthesis [151].


### 3.3. Findings in Clinical Studies

A substantial body of clinical research has established a robust link between the gut microbiota and depression. Multiple studies have consistently demonstrated significant differences in the composition of the gut microbiota between individuals with major depressive disorder (MDD) and healthy controls [6,152,153,154,155,156,157,158]. At the phylum level, compared with healthy controls, patients with MDD typically present a reduced proportion of *Firmicutes* and increased proportions of *Bacteroidetes*, *Proteobacteria*, and *Actinobacteria* [152,153,154]. At the family level, *Bacteroidaceae* (MDD: 27.79% vs. healthy controls: 26.93%) and *Prevotellaceae* (MDD: 15.99% vs. healthy controls: 14.95%) were found to be more abundant in MDD patients [155,156]. Conversely, families such as *Prevotellaceae* (MDD: 7.75% vs. healthy controls: 9.91%), *Bifidobacteriaceae* (MDD: 3.91% vs. healthy controls: 4.22%), *Ruminococcaceae* (MDD: 16.78% vs. healthy controls: 17.04%), *Selenomonadaceae* (MDD: 4.07% vs. healthy controls: 5.20%), and *Enterobacteriaceae* (MDD: 3.78% vs. healthy controls: 5.26%) presented decreased relative abundances [155,156]. At the genus level, several genera, including *Bilophila*, *Alistipes* [157], *Eggerthella*, *Gemella*, *Holdemania*, *Turicibacter* and *Paraprevotella* [6,158] were found to be more abundant in MDD patients than in healthy controls. In contrast, genera such as *Anaerostipes*, *Dialister* [157], *Prevotella* [158], *Butyricicoccus*, *Coprococcus, Faecalibacterium*, *Eubacterium_ventriosum_group*, *Fusicatenibacter*, *Romboutsia*, and *Subdoligranulum* [6] presented reduced relative abundances.

Subsequent investigations have explored the relationship between specific gut bacteria and depression, revealing that certain bacteria possess antidepressant potential. Over the past two decades, a new class of probiotics known as psychoprobiotics or psychomicrobes (e.g., *Lactobacilli*, *Streptococci*, *Bifidobacteria*, *Escherichia* and *Enterococci*) has emerged as potential interventions for various mental disorders [159]. A systematic review of 15 clinical trials conducted between 2004 and 2025 (Table 2) indicated that most studies reported that probiotics had positive effects on measures of depressive symptoms. Additionally, some clinical trials have shown that administering probiotics containing specific strains of *Bifidobacteria* and *Lactobacilli* to MDD patients over an 8–10-week period resulted in significant reductions in self-rated depression scores and improvements in associated symptoms such as anxiety [160,161,162,163,164]. However, owing to variations in clinical trial design, including differences in probiotic dose, strain selection, and treatment duration, further large-scale randomized controlled trials are necessary to validate the efficacy and safety of probiotics in treating depression.

## 4. Correlations Between the Plateau Environment, the Gut Microbiota, and Depression

### 4.1. Influence of the Plateau Environment on Depression via the Gut Microbiota

In Section 2, we detail the unique characteristics of the plateau environment, including low oxygen levels, low temperatures, and high radiation, all of which significantly impact the human gut microbiota. Changes in the gut microbiota are closely associated with the onset and progression of depression [111]. This may explain why the prevalence of depression is greater in plateau regions than in plain regions [11,14,17]. These three factors—the plateau environment, the gut microbiota, and depression—interact through the “plateau environment-gut microbiota-depression” axis. The harsh conditions of hypoxia, low-temperature, and high radiation in plateau regions disrupt the homeostasis of the gut microbiota, triggering a series of physiological chain reactions. Once the gut microbiota becomes imbalanced, it communicates with the brain via the gut-brain axis, influencing key processes such as neurotransmitter synthesis and release, immune cell activation and regulation, and metabolite generation and transport. These changes ultimately contribute to the development of depressive symptoms [111]. In the following sections, we explore the relationships among the plateau environment, the gut microbiota, and depression from three perspectives: neural conduction, immune regulation, and metabolic pathways.

#### 4.1.1. Neurotransmitter Conduction

According to the findings in Section 3.2.1, plateau environments significantly intensify gut microbiota dysregulation, as evidenced by reduced levels of beneficial genera such as *Bifidobacterium* and *Lactobacillus* [18,47]. This disruption adversely affects tryptophan metabolism and serotonin (5-HT) synthesis. Such changes parallel those observed in depression-associated microbiota alterations [155,156], where diminished serotonin availability plays a critical role in contributing to mood disturbances and heightened anxiety [148,162,163]. On the basis of this evidence, we hypothesize that the plateau environment may increase the incidence of depression by disrupting specific gut microbiota and consequently affecting 5-HT synthesis. However, further clinical studies are needed to confirm the critical role of 5-HT in altitude-induced depression.

#### 4.1.2. Immune Regulation

As detailed in Section 3.2.2, beneficial bacteria (e.g., *Bifidobacterium*) maintain gut barrier integrity via tight junctions and SCFA production. From an immunological perspective, environmental changes at high altitudes can disrupt microbial balance, leading to impaired gut barrier function. This allows bacterial translocation into the bloodstream, activating peripheral immune responses and increasing the release of inflammatory cytokines such as IL-6 and TNF-α. These cytokines cross the blood-brain barrier, inducing CNS inflammation, damaging neurons, and affecting neuroplasticity, potentially contributing to high-altitude depression [121,122,166].

#### 4.1.3. Metabolic Pathways

The metabolic pathways of the gut microbiota play crucial roles in maintaining health. The production of SCFAs by the gut microbiota is altered by high-altitude environments. For example, reduced levels of butyric acid (an important SCFA) weaken the gut barrier via downregulation of Claudin-1 and inhibit microglial M2 polarization via HDAC6 inactivation [145], leading to increased tryptophan catabolism via IDO activation and HPA axis hyperactivity via GR promoter hypomethylation. In particular, overactivation of the HPA axis leads to abnormal cortisol secretion, enhances the stress response, and exacerbates depression. Clinical studies support this hypothesis; fecal sample analyses from depressed patients in high-altitude regions revealed significantly reduced microbial diversity, a decreased abundance of SCFA-producing bacteria, and increased plasma inflammatory cytokine levels, and these changes were positively correlated with the severity of depression (Figure 3). These findings suggest a potential role for the high-altitude environment in mediating depression through gut microbiota alterations, but direct evidence of this pathway remains limited.

The role of gut microbiota metabolic pathways in altitude-induced depression has emerged as a potential key link. Under hypoxic, low-temperature, and high radiation conditions, studies have revealed decreases in the abundances of gut bacteria such as *Lactobacillus*, *Bifidobacterium*, and *Ruminococcaceae* in human and mouse models [18,46,59,60]. These changes affect SCFA synthesis, particularly by reducing butyric acid levels [145,167,168]. Interestingly, butyrate supplementation increases *Lactobacillaceae* populations while decreasing *Actinomyceaceae* and *Dantobaccoaceae* populations [48], highlighting the close relationship between butyrate and the gut microbiota in high-altitude environments. Butyric acid, a key SCFA, plays a critical role in maintaining tight junction protein expression, ensuring gut barrier integrity. Its deficiency leads to increased neuroinflammation via microglial activation and proinflammatory cytokine release [148]. This disorder exacerbates neurotransmitter imbalances and disrupts the HPA axis, resulting in sustained high cortisol secretion and amplified stress responses [146]. In light of these findings, butyrate has emerged as a promising therapeutic target for high-altitude depression.

#### 4.1.4. Interaction Mechanism of Multiple Environmental Factors

The hypoxic environment disrupts the anaerobic-aerobic balance of the gut microbiota by inhibiting the growth of aerobes (such as *E. coli*) while promoting the overproliferation of anaerobes (such as *Bacteroides*) [39]. This alteration in microbial community structure directly affects the stability of the gut microecology. The low-temperature environment further inhibits the metabolic activity of probiotics (such as *Lactobacillus*), significantly reducing the production of SCFAs [48]. As a key substance maintaining gut barrier function, a reduction in SCFAs weakens the protective effect of the gut barrier. The high radiation environment induces gut oxidative stress, leading to a significant decline in the number of beneficial bacteria (such as *Akkermansia*) and a large-scale enrichment of pathogenic bacteria (such as *Clostridium*) [60], which not only exacerbates dysbiosis but also directly damages the gut mucosa due to oxidative stress. The synergistic effect of these three environmental factors leads to more severe consequences. The microbial imbalance is associated with hypoxia, a reduction in SCFAs due to low-temperature, and gut mucosal damage and dysbiosis resulting from high radiation superimposed on each other, greatly exacerbating gut barrier damage. Disruption of the gut barrier makes bacterial metabolites (such as LPS) more likely to enter the bloodstream, activating a systemic inflammatory response. These inflammatory factors affect key processes such as neurotransmitter synthesis and release, immune cell activation and regulation, and metabolite production and transport through the gut–brain axis, forming a vicious cycle that ultimately promotes the generation of depressive symptoms. This interaction mechanism of multiple environmental factors is complex, involving changes in multiple links and levels, jointly promoting the association between the gut microbiota and depression in the plateau environment, and highlighting the complexity and integrity of the synergistic effects of multiple factors in the plateau environment on the gut microbiota and the occurrence and development of depression.


**A potential core mechanism by which the high-altitude environment may induce depression via the gut microbiota could involve multipathway synergistic dysregulation, though this remains speculative and requires validation (Figure 4):**


(1)**Impaired nerve conduction:** Hypoxia, low-temperature, and radiation result in a reduced abundance of beneficial bacteria such as *Bifidobacterium* and *Lactobacillus*. This inhibits tryptophan metabolism and 5-HT synthesis, thereby disrupting mood regulation pathways.(2)**Immune-inflammatory activation:** Dysbiosis compromises intestinal barrier function, leading to endotoxin translocation. This activates systemic inflammation (e.g., elevated IL-6 and TNF-α) and induces neuroinflammation and neuroplasticity damage through the blood–brain barrier.(3)**Metabolic homeostasis imbalance:** Reduced production of SCFAs, particularly butyric acid, weakens intestinal barrier protection and anti-inflammatory capacity. Concurrently, excessive activation of the HPA axis (e.g., elevated cortisol levels) results in a vicious cycle of “stress–inflammation.”(4)**Multifactorial synergistic toxicity:** Hypoxia promotes an anaerobic/aerobic imbalance in the microbiota, low-temperature suppresses probiotic metabolism, and radiation induces oxidative stress. The cumulative effects of these factors exacerbate microbial dysregulation and amplify depression risk through the “microbiota-inflammation-brain” axis.(5)**Core points:** Multiple stressors in high-altitude environments trigger the key pathological chain of depression via the “neuro-immune-metabolite” three-dimensional regulatory network of the gut microbiota. Interventions targeting bacterial metabolites (e.g., butyric acid) or probiotics hold promise as precise prevention and control strategies for high-altitude depression.

#### 4.1.5. Limitations

Notably, the majority of studies associating plateau environments with gut microbiota–depression pathways are cross-sectional or correlational in nature. Direct causal evidence, such as longitudinal studies tracking microbial changes concurrent with the onset of depression in plateau populations or gnotobiotic animal models simulating altitude-induced stress, remains scarce. For instance, although Table 1 indicates higher rates of depression among plateau populations, these data lack concurrent profiling of the gut microbiota. Future research should adopt prospective cohort designs or intervention studies, such as fecal microbiota transplantation in altitude-related models, to establish and validate causal relationships.

### 4.2. Population-Specific Differences in the Gut Microbiota-Immune-Neuroendocrine Axis in High-Altitude Environments

#### 4.2.1. Populations Exposed Acutely to High Altitudes

Acute exposure to high altitudes (<1 week) induces significant and dynamic alterations in the gut microbiota, characterized by an overgrowth of aerobic bacteria (e.g., *Escherichia coli*) and a marked reduction in anaerobic bacteria (e.g., *Blautia*, *Lachnospiracea incertae sedis*, and *Bifidobacterium*) (Table 3). These microbial changes are associated with SCFA and carbohydrate metabolism [18]. An animal study demonstrated that exposing rats to a simulated hypoxic environment at 4000 m for three days resulted in gut barrier damage [169]. Notably, hypoxia-inducible factor 1-alpha (HIF-1α) is significantly upregulated in the gut mucosa, which may contribute to gut barrier dysfunction [41]. Consequently, this microbial imbalance may be closely linked to the development of acute mountain sickness (AMS), characterized by gut barrier dysfunction [170]. While AMS involves physiological stress, its direct relationship to psychological stress and depression requires further validation. A 16-year clinical cohort study in Taiwan, China, revealed that compared with controls, subjects who developed AMS had a nearly tenfold increased risk of developing psychiatric disorders [172]. Additionally, the susceptibility of AMS patients to psychiatric disorders may be associated with low-grade inflammatory responses and elevated cortisol levels [173]. As discussed in Section 4.1.2 and Section 4.1.3, the gut microbiota plays a crucial role in regulating inflammation and the HPA axis, and the microbial changes induced by acute altitude exposure are also implicated in these regulatory pathways.

#### 4.2.2. Chronic Plateau Adaptation Population

Long-term plateau dwellers have developed unique gut microbiota adaptation characteristics. For example, a comparative analysis of the gut microbiota of Tibetan and Han Chinese populations at low and high altitudes (Nyingchi, Tibet, average altitude 3100 m) revealed that, at the phylum level, the relative abundance of *Firmicutes* was relatively high, whereas the relative abundance of *Bacteroidetes* was relatively low in both Tibetan and Han Plateau flora. At the genus level, *Actinomyces*, *Blautia*, *Clostridium*, *Desulfovibrio*, *Helicobacter*, *Leuconostoc*, *Peptostreptococcaceae Incertae Sedis*, and *Rhodococcus* presented relatively high abundances (Table 3) [147]. Das et al. also confirmed the relatively high abundance of *Firmicutes* and *Prevotella* in an analysis of the flora of populations residing in high-altitude regions of India [174]. Interestingly, owing to distinct dietary patterns in plateau regions (Tibetans prefer ghee and consume substantial amounts of beef and mutton products with minimal intake of vegetables and fruits), the abundance of *Collinsella* was significantly elevated in individuals using ghee as cooking oil [174]. *Collinsella* has been associated with high serum cholesterol levels [175]. Additionally, some studies have demonstrated that low-pressure and hypoxic conditions in plateau areas can induce inflammatory responses in mammals, leading to vascular leakage and the accumulation of inflammatory cells in multiple organs [176,177,178].

Notably, chronic hypoxia, as the central stressor of the plateau environment, induces neurocognitive dysfunction through multidimensional pathological mechanisms [179,180]. Maiti et al. reported that rats exposed to a simulated altitude of 6100 m for 21 days presented significant regional differences in response to oxidative stress: the activity of superoxide dismutase (SOD) in the hippocampus (a core region of cognitive function) decreased most markedly, and the compensatory increase in glutathione peroxidase (GPx) activity was the shortest duration; the level of 8-OHdG, a marker of oxidative damage in the prefrontal cortex, was negatively correlated with spatial memory ability. Compared with other brain regions, striatal dopaminergic neurons are more sensitive to oxidative stress and exhibit a faster decline in the mitochondrial membrane potential [181]. Similarly, human studies have shown that long-term residence at high altitudes leads to a persistent imbalance between oxygen free radical formation and antioxidant defenses, resulting in systemic oxidative nitration, inflammatory stress, and accelerated cognitive impairment in patients with chronic altitude sickness. Furthermore, hypoxemia-induced systemic oxidation-inflammation-nitrite stress in plateau residents represents a physiological continuum, which, when overactivated, is associated with accelerated cognitive decline and depression [182]. Recent investigations have indicated that high-altitude exposure may also cause mitochondrial dysfunction in the brain, thereby increasing the risk of oxidative stress and bipolar disorder [183]. Importantly, altitude differences are linked to depression, and hypobaric hypoxia may contribute to suicide and depression via its association with serotonin metabolism and brain bioenergetics [184].

These findings regarding the relationships among altitude-related depression, inflammation, and TRP metabolism suggest that the gut microbiota has important functional potential. However, mammals native to plateau regions (plateau rats and plateau pikas) that have adapted to such environments over extended periods can utilize flora (*Lachnospiraceae* and *Clostridiaceae*) to produce large quantities of SCFAs [185,186], which resist inflammation by inhibiting the nuclear factor-kappa B (NF-κB) pathway or histone deacetylase function [145]. A similar phenomenon was observed in a study where Li et al. analyzed stool samples from humans and Tibetan pigs living in plateau areas, revealing that both the human and Tibetan gut microbiota may produce greater amounts of SCFAs [148]. Notably, increasing dietary intake of plant fiber can reduce inflammation by increasing microbial diversity and aid in accelerating the body’s adaptation to the plateau environment [187], potentially providing insights into interventions for high-altitude depression.

#### 4.2.3. Compound Risk in Special Populations

Special populations, such as pregnant women and elderly individuals residing in plateau regions, face a more complex interplay between the plateau environment, the gut microbiota, and mental health conditions such as depression. Pregnant women at high altitudes encounter the dual challenges of increased physiological stress and environmental factors during pregnancy. Hormonal fluctuations during pregnancy can disrupt gut permeability [188] and microbial homeostasis [189], whereas hypoxia and elevated radiation levels may exacerbate these effects. A reduction in beneficial bacteria, such as gut *Bifidobacteria*, not only impairs maternal nutrient absorption but also affects fetal neurodevelopment through vertical transmission from mother to child, thereby increasing the risk of postpartum depression in offspring [190]. In elderly individuals living in plateau areas, the diversity of the gut microbiome significantly decreases with age, leading to an increased risk of depression as gut mucosal barrier function declines [190,191,192]. A high-altitude environment may accelerate this process, thereby influencing the incidence of depression. However, owing to insufficient clinical and foundational research, a comprehensive review of the specific risks faced by special populations in plateau regions remains challenging.

#### 4.2.4. Limitations

**The observed associations must be interpreted alongside potential confounders.** For example, sleep disorders—prevalent in high-altitude environments owing to hypoxia-induced breathing disruptions [193]—are independently linked to both gut dysbiosis and depression. Additionally, genetic factors (e.g., hypoxia-inducible factor polymorphisms in Tibetan populations), dietary patterns (high fat/low fiber intake), and social stressors (isolation, limited mental health access) may confound the proposed microbiota–brain axis. Future studies should employ multivariate regression to control for these variables and use twin designs to disentangle genetic and environmental effects.**Comorbidity and socioeconomic factors:** This review also acknowledges limitations in addressing comorbidities (e.g., chronic mountain sickness, pulmonary hypertension) and socioeconomic factors (e.g., limited mental health access, cultural stigma), which may confound the microbiota-depression relationship in plateau populations. Future studies should adopt multidisciplinary approaches to disentangle these interacting variables.**Genetic adaptation in high-altitude populations,** such as Tibetans, may influence the gut microbiota composition and depression risk. For example, Tibetans exhibit unique genetic variants in the HIF-1α and EPAS1 pathways, which regulate hypoxia tolerance and may interact with gut bacteria to modulate SCFA production [194]. Additionally, dietary patterns in plateau regions—characterized by high intake of ghee, red meat, and low fiber—promote the growth of *Collinsella* and reduce SCFA-producing bacteria such as *Ruminococcaceae*. These dietary habits may exacerbate gut dysbiosis and inflammation, independent of altitude stress.

## 5. Intervention Strategies: From Theory to Practice

### 5.1. Potential Methods for Improving Depression in Plateau Environments Through the Regulation of the Gut microbiota

The interventions discussed here (dietary fiber, probiotics, FMT) complement established first-line treatments for depression (e.g., cognitive behavioral therapy, exercise) [195,196,197,198] and are proposed as adjunctive strategies for plateau populations with unique microbial dysregulation. For example, while CBT remains the gold standard for psychological intervention, probiotics may offer the following advantages: (1) targeting gut–brain pathways disrupted by altitude stress; (2) feasibility in resource-limited settings; and (3) minimal side effects. Table 2 summarizes the evidence hierarchy for these interventions, prioritizing probiotics (level B evidence from RCTs) over FMT (level C, experimental).

(1)Dietary Fiber

Given our previous discussion, the gut microbiota plays a pivotal role in altitude-induced depression. Consequently, targeted modulation of the gut microbiome represents a promising approach to enhance mental health among plateau populations. Nutritional interventions via diet serve as a foundational and readily implementable means. Increasing dietary fiber intake can stimulate the proliferation of beneficial bacteria and increase SCFA levels [199]. Residents in plateau regions are advised to moderately increase their consumption of barley, oats, and other grains rich in dietary fiber, along with a variety of locally available fresh vegetables and fruits. Most countries recommend a daily intake of 25–30 g of dietary fiber, which exceeds the global average of 15–26 g per day [200]. A clinical study demonstrated that direct supplementation with inulin over three weeks increased the relative abundance of *Bifidobacteria* from 6.69% to 15.07% [201]. Another study indicated that after two weeks of supplementation with inulin-rich vegetables, *Bifidobacteria* levels increased 3.8-fold, leading to a slight improvement in intrinsic emotional capacity [202]. Therefore, inulin supplementation can selectively increase the proliferation of *Bifidobacteria*, thereby improving the gut microecology. Concurrently, this is associated with a reduction in the levels of serum inflammatory markers [201], such as IL-6, among subjects, which implies that the antidepressant effects of dietary fiber may be linked to microbiota-immune regulation.

(2)Probiotics

Certain classes within the gut microbiota are generally recognized as beneficial and are referred to as probiotics. When administered in adequate amounts, these microorganisms confer health benefits to the host. The two most commonly studied strains are *Lactobacillus* and *Bifidobacterium* [203,204]. These probiotic strains can enhance the gut mucosal barrier, mitigate inflammation and oxidative stress, and alleviate depression symptoms induced by high-altitude environments through the production of SCFAs, particularly butyrate [205]. Notably, in high-altitude regions, Tibetans have a dietary preference for dairy products, especially yogurt, which is rich in *Lactobacilli*. This dietary habit may contribute to their adaptation to the plateau environment and help mitigate AMS symptoms [206]. Animal studies have demonstrated that supplementation with *Bifidobacterium longum JBLC-141* can restore gut barrier function in rats exposed to simulated high-altitude conditions (elevation: 6000 m; oxygen content: 13%; air pressure: 354 mmHg). This intervention reduces inflammatory responses and upregulates the expression of tight junction proteins [207]. Furthermore, a study on Chinese military personnel at high altitudes revealed that continuous administration of probiotics could prevent and treat gastrointestinal stress reactions and dysbiosis [207]. Therefore, incorporating probiotics into foods and medications may offer preventive and therapeutic benefits for high-altitude-induced depression.

(3)Fecal Microbiota Transplantation

As an emerging therapeutic strategy, FMT has demonstrated significant potential. This process involves the transplantation of the fecal microbiota from carefully screened healthy donors into the guts of patients to restore the gut microecological balance [201]. In several clinical and animal studies [208,209,210,211,212], patients with treatment-resistant depression who received FMT exhibited rapid recovery of gut microbiota diversity, normalization of previously disrupted metabolic functions, and marked alleviation of depressive symptoms, including anhedonia and cognitive impairments. Some patients were even able to resume normal daily activities and work. Consequently, FMT holds promise as a potential intervention for patients in the plateau phase with severe gut dysbiosis and refractory depressive symptoms. However, FMT remains in the experimental stage. Rigorous protocols are needed for donor screening, determination of transplantation dosage, and selection of transplantation routes to ensure safety and efficacy. Large-scale, long-term clinical trials are necessary to further validate its sustained effectiveness and assess any potential risks [213].

### 5.2. Other Supporting Interventions

(1)Psychological Intervention

Psychological intervention plays an indispensable role in alleviating depressive symptoms [195]. Cognitive behavioral therapy (CBT) is a widely recognized approach for treating psychiatric disorders, including depression and anxiety. By assisting patients in identifying and correcting negative thought patterns and behaviors, CBT facilitates the reconstruction of positive cognition [196]. In plateau regions, individuals with depression frequently experience negative thoughts such as “I cannot adapt to this environment” because of physical discomfort and harsh environmental conditions. CBT therapists can guide patients to perceive physiological reactions as a normal part of altitude acclimatization, encourage gradual engagement in social activities, and promote participation in moderate cultural and recreational activities, thereby breaking self-imposed isolation. Drawing from the psychological intervention timelines for depression in non-altitude areas, most patients undergo systematic treatment lasting 1–12 months, during which their depressive and anxious symptoms significantly diminish, and their confidence in coping with life markedly improves [196]. Supportive psychotherapy provides emotional catharsis channels for patients; therapists patiently listen to the work-related, lifestyle pressures, and distress experienced by individuals in plateau environments, offering understanding, encouragement, and practical advice. Regular psychological support serves as a “spiritual harbor,” alleviating loneliness and helplessness, enhancing psychological resilience, and preventing the exacerbation of depression.

(2)Exercise

Research has demonstrated that exercise effectively mitigates depressive symptoms [197,198]. Schmitter et al. [214] conducted a 2020 clinical study revealing that home-based exercise 1–2 times per week alleviates depression among patients. Another study [215] revealed that a 12-week yoga intervention in individuals with major depression increased GABA levels and reduced depressive symptoms. For women with postpartum depression, a 4-week exercise program increases serotonin levels in the brain, reduces the severity of postpartum depressive symptoms, and alleviates anxiety [216]. Additionally, exercise improves altitude fitness [217], and appropriate long/short sprint tests increase erythropoietin concentrations and red blood cell counts, thereby increasing aerobic capacity [218]. Consequently, selecting an appropriate exercise regimen promotes altitude adaptability and reduces the incidence of depression. For example, aerobic exercise, such as jogging, brisk walking, and cycling, effectively promotes blood circulation, enhances the body’s tolerance to hypoxic plateau environments [218], reduces anxiety, improves mental focus, assists individuals in better managing physical and psychological discomfort associated with environmental stressors, enhances quality of life, and decreases the prevalence of depression.


**The depression intervention strategy targeting the intestinal microbiota in the plateau environment can be summarized as a comprehensive plan with “microbiota regulation as the primary approach and multimodal support as an auxiliary component”.**


(1)**Dietary Fiber Intervention:** Consuming foods rich in dietary fiber, such as oats and barley, selectively promotes the proliferation of *Bifidobacteria* and *Lactobacillus*. This increases the levels of SCFAs (e.g., inulin supplementation for three weeks can increase *Bifidobacteria* abundance), strengthens the intestinal barrier, suppresses inflammation, and improves mood regulation function.(2)**Probiotic application:** Supplementing probiotics, such as *Lactobacillus* and *Bifidobacterium*, repairs intestinal mucosal barrier damage caused by hypoxia at high altitudes (e.g., *Bifidobacterium longifolium JBLC-141* upregulates tight junction protein expression). It reduces proinflammatory factor levels, and the daily intake of probiotic-rich yogurt by the Tibetan population may correlate with their adaptability to high-altitude conditions.(3)**FMT:** As an emerging therapeutic approach, FMT rapidly restores intestinal diversity in patients with severe microbiota imbalance and alleviates symptoms of refractory depression. However, overcoming technical challenges, such as donor screening and dose standardization, is necessary. Currently, FMT remains in the stages of animal experimentation and clinical exploration.(4)**Supportive interventions:** CBT reshaped negative cognition to alleviate high-intensity adaptation stress. Regular exercise (e.g., 1–2 aerobic sessions per week) increases serum serotonin levels and enhances hypoxia tolerance. The integration of CBT and exercise with microbiota regulation further mitigates the risk of depression through neuroimmune-metabolic pathways.(5)**Core Points:** Flora regulation based on dietary fiber and probiotics constitutes the core strategy for addressing high-altitude depression. The FMT demonstrates potential breakthrough value. Psychological and exercise interventions play synergistic roles via neuro-immune-metabolic pathways, collectively constructing a multilevel prevention and control system.

## 6. Conclusions and Prospects

The plateau environment, characterized by hypoxia, low-temperature, and high radiation, induces depressive symptoms through gut microbiota dysregulation via the gut-brain axis. Key mechanisms include reduced SCFA production, neurotransmitter imbalance (e.g., serotonin depletion), and immune-inflammatory activation. Clinical evidence highlights elevated depression prevalence in plateau populations, which is linked to microbial shifts such as *Firmicutes* enrichment and SCFA metabolic abnormalities. However, research gaps persist, including unclear causal relationships between environmental stressors and microbial function, limited population-specific data (e.g., altitude gradients, genetic backgrounds), and insufficient clinical validation of interventions such as probiotics and FMT. Notably, a significant limitation of this review is the absence of a quantitative meta-analysis, which hinders the ability to estimate pooled effect sizes and synthesize consistency across studies. For example, while existing studies report associations between gut microbiota alterations (e.g., reduced *Firmicutes*/SCFAs) and depression in plateau populations, the lack of meta-analytic integration prevents robust conclusions about the generalizability of microbial signatures (e.g., decreased butyrate-producing bacteria) or the efficacy of interventions. Future research should adopt systematic review protocols (e.g., PRISMA) to conduct meta-analyses on high-altitude cohorts, quantifying the strength of associations between microbial dysbiosis and depression. This approach could clarify whether microbial patterns (e.g., the *Firmicutes*/*Bacteroidetes* ratio and SCFA levels) are consistent across diverse plateau populations and validate their utility as depression risk biomarkers. Concurrently, integrating multiomics (metagenomics, metabolomics) and single-cell sequencing could decode the “environment-microbiota-brain” regulatory network, whereas AI-driven big data analysis and gut organoid models may facilitate the development of precision therapies, such as customized probiotics targeting SCFA pathways or histone deacetylase activity. By combining meta-analytic rigor with cutting-edge omics technologies, this interdisciplinary framework will not only address current limitations but also enhance the translational potential of “microbiota-brain” research, providing evidence-based strategies for mental health protection in high-altitude environments and deepening our understanding of extreme environment-neuropsychiatric interactions.

## Figures and Tables

**Figure 1 cimb-47-00487-f001:**
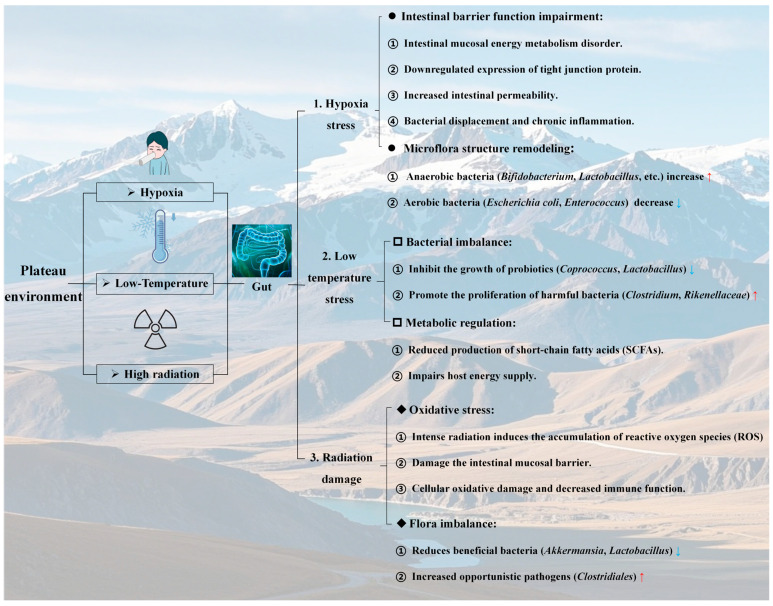
Mechanistic schematic of the effects of high-altitude environmental factors on the gut. Note: This diagram illustrates the mechanisms by which three high-altitude environmental factors—hypoxia, low-temperature, and high radiation—act on the gut. (1) Hypoxic stress: ① The microbial structure of anaerobes (e.g., *Bifidobacterium*, *Lactobacillus*) can be remodeled, and aerobes (e.g., *Escherichia coli*, *Enterococcus*) can be decreased. ② Impairing gut barrier function can induce gut mucosal energy metabolism disorders, downregulate tight junction protein expression, increase gut permeability, promote bacterial translocation, and trigger chronic inflammation. (2) Low-temperature stress: ① Inhibits the growth of probiotics (e.g., *Pseudomonas cocovenenans*, *Lactobacillus*) and promotes the proliferation of harmful bacteria (e.g., *Clostridium*, *Rikenellaceae*), leading to bacterial dysbiosis; ② reduces the production of short-chain fatty acids (SCFAs), impairs the host energy supply, and disrupts metabolic regulation. (3) Radiation damage: ① Intense radiation induces reactive oxygen species (ROS) accumulation, causing oxidative stress; ② Disrupts the gut mucosal barrier, leading to cellular oxidative damage and immunocompromise; ③ Reduces beneficial bacteria (e.g., *Akkermansia* and *Lactobacillus*) and increases opportunistic pathogens (e.g., *Clostridiales*), resulting in microbial dysbiosis. (The red arrow signifies an upward trend, whereas the blue arrow denotes a downward trend.)

**Figure 2 cimb-47-00487-f002:**
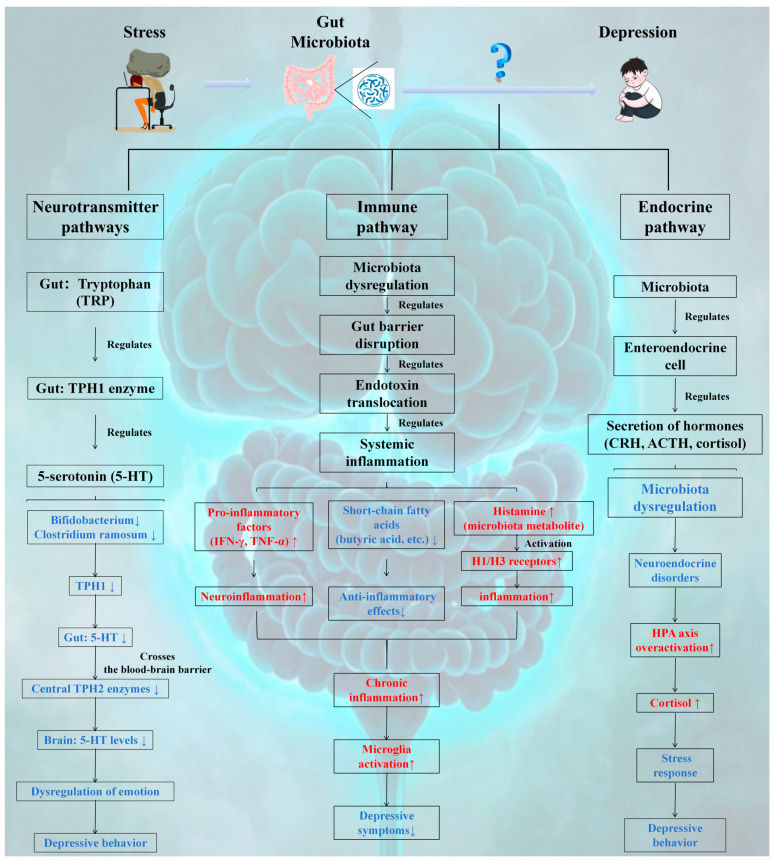
Mechanisms by which the gut microbiota mediates stress and depression: Neurotransmitter, Immune, and Endocrine Pathways. Note: The figure illustrates the mechanism of action of the gut microbiota between stress and depression, which is mediated by three key pathways: neurotransmitter, immune, and endocrine. (1) Neurotransmitter pathway: TRP in the gut is involved in the synthesis of 5-HT under the action of the TPH1 enzyme. *Bifidobacterium*, *Clostridium ramosum* and others regulate TPH1. Intestinal 5-HT affects the central TPH2 enzyme through the blood–brain barrier, ultimately influencing the level of 5-HT in the brain and regulating emotions, and abnormalities in this protein can lead to depressive behavior. (2) Immune pathway: Dysbiosis triggers intestinal barrier disruption, and endotoxin translocation leads to systemic inflammation. Proinflammatory factors (such as IFN-γ and TNF-α) induce neuroinflammation, whereas anti-inflammatory substances such as SCFAs (such as butyric acid) exert anti-inflammatory effects. Histamine (a microbial metabolite) activates H1/H3 receptors, exacerbating inflammation. Chronic inflammation activates microglia, ultimately leading to depressive symptoms. (3) Endocrine pathway: Gut microbes regulate enteroendocrine cells, affecting the secretion of hormones such as CRH, ACTH, and cortisol. Dysbiosis leads to neuroendocrine disorders, overactivation of the HPA axis, increased cortisol levels, triggering a stress response, and ultimately leading to depressive behavior. The figure clearly shows the interactions of key substances and links in each pathway, revealing the complex mediating mechanisms of the gut microbiota in the stress–depression association. (The red arrow signifies an upward trend, whereas the blue arrow denotes a downward trend.)

**Figure 3 cimb-47-00487-f003:**
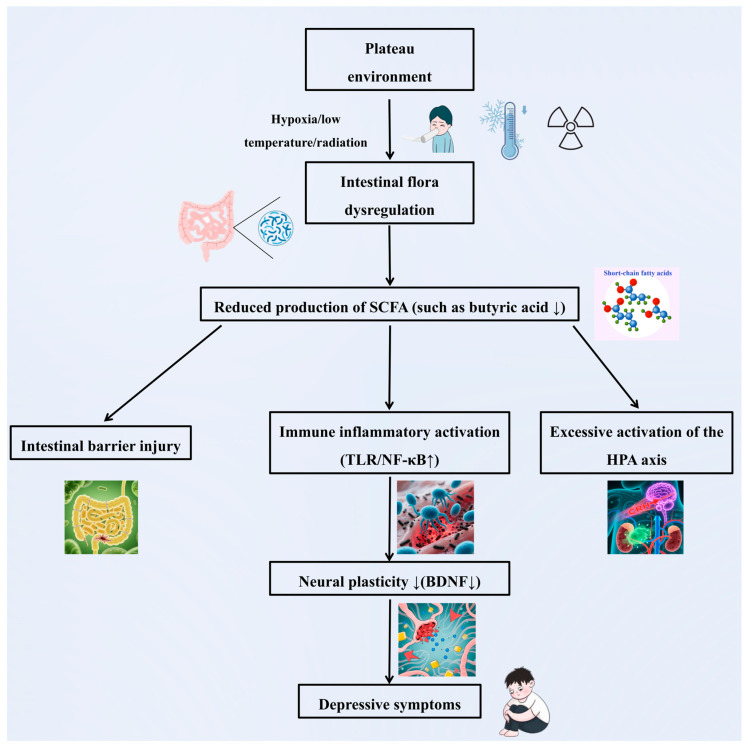
Central component of the SCFA molecular pathway. Note: (1) Impaired synthesis: A high-altitude environment suppresses the activity of short-chain fatty acid (SCFA)-producing bacteria, such as *Firmicutes*, resulting in reduced levels of butyric acid and propionic acid; (2) weakened barrier protection: SCFAs maintain the expression of tight junction proteins (e.g., ZO-1 and Claudin-1) via the GPR43 receptor. Their deficiency directly contributes to intestinal permeability and barrier dysfunction. (3) Immune-neural interaction disorder: Butyric acid inhibits the NF-κB signaling pathway, thereby reducing the secretion of the proinflammatory cytokines IL-6 and TNF-α. Propionic acid activates the GPR43 receptor in microglia, promoting M1-type polarization and contributing to neuroinflammation. (4) Uncontrolled HPA axis: SCFAs regulate the methylation of the CRH promoter through histone deacetylases (HDACs). Their insufficiency impairs cortisol feedback inhibition, leading to dysregulation of the hypothalamic-pituitary-adrenal (HPA) axis.

**Figure 4 cimb-47-00487-f004:**
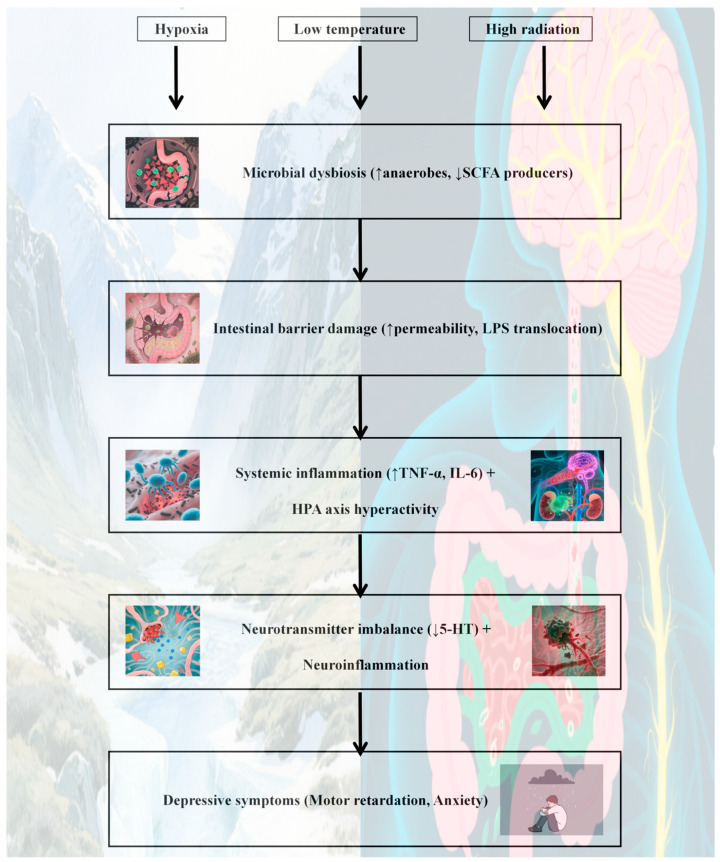
Synergistic mechanism of the three factors in the plateau environment.

**Table 2 cimb-47-00487-t002:** Probiotic intervention.

Years	Subject Population	Microbiota	Intervention Duration	Effect
2002 [163]	Adults suffering from stress or exhaustion (*n* = 34)	*Lactobacillus acidophilus*, *Bifidobacterium bifidum* and *Bifidobacterium longum*	6 months	The subjects’ stress improved by 40.7% overall
2004 [164]	College students under exam pressure (*n* = 136)	*Lactobacillus casei*, *Lactobacillus delbrueckii subspecies bulgaricus* and *Streptococcus salivarius subspecies thermophiles*	6 weeks	Increased the number of lymphocytes and CD56 cells in subjects
2009 [148]	Chronic Fatigue Syndrome (CFS) patients (*n* = 35)	*Lactobacillus casei strain Shirota*	8 weeks	The anxiety scores of CFS patients were significantly reduced
2010 [165]	Irritable Bowel Syndrome (IBS) patients (*n* = 74)	*Lactobacillus paracasei*, *ssp. paracasei F19*, *Lactobacillus acidophilus La5* and *Bifidobacterium lactis Bb12*	8 weeks	Reduce Anxiety and Depression (HAD) scale scores in IBS subjects
2011 [166]	Subjects with urine free cortisol (UFC) levels below 50 ng/mL (*n* = 25)	*Lactobacillus helveticus R0052* and *Bifidobacterium longum R0175*	30 days	Significantly reduced HAD scores in UFS subjects
2013 [167]	College students who exercise vigorously every day (*n* = 44)	*Lactobacillus gasseri OLL2809*	4 weeks	The effects of increasing the activity of natural killer cells reduced by intense exercise in subjects and improving mood in depressed states may help athletes maintain physical and mental health
2015 [168]	Adults with IBS and diarrhea or mixed stool patterns (based on the Rome III criteria) and mild to moderate anxiety and/or depression (based on the Hospital Anxiety and Depression Scale) (*n* = 44)	*Bifidobacterium longum NCC3001*	6 weeks	HAD scores were significantly reduced, and negative emotional stimulus responses in the amygdala and frontal limbic regions were reduced.
2016 [145]	Laryngeal Cancer (LC) patients (*n* = 20)	*Clostridium butyricum*	2 weeks	After taking Cb, the anxiety level of LC patients was relieved, and the serum adrenocorticotropin-releasing factor was inhibited
2016 [146]	MDD patients (*n* = 40)	*Lactobacillus acidophilus*, *Lactobacillus casei* and *Bifidobacterium bifidum*	8 weeks	The total score of the Baker Depression Scale (BDI) was significantly reduced
2018 [169]	Patients with moderate depression (*n* = 40)	*Bifidobacterium breve*, *Bifidobacterium longum*, *Lactobacillus acidofilus*, *Lactobacillus bulgarigus*, *Lactobacillus casaei*, *Lactobacillus rhamnosus* and *Streptococus thermophilus*	10 weeks	Patients with MD had significantly lower Hamilton Depression Scale (HAM-D) scores
2019 [170]	Patients with clinically diagnosed depression (*n* = 110)	*Lactobacillus helveticus* and *Bifidobacterium longum*	8 weeks	Patients with depression had significantly lower total BDI scores and significantly lower serum kynurenine/tryptophan ratios
2021 [171]	MDD patients (*n* = 10)	*Lactobacillus helveticus R0052* and *Bifidobacterium longum R0175*	6 weeks	Overall early depressive symptoms improved significantly in MDD patients as measured by MADRS and QIDS-SR16, as well as anhedonia-like symptoms as measured by SHAPS.
2022 [172]	Patients with mild to moderate depression (*n* = 47)	*Streptococcus thermophilus NCIMB 30438*, *Bifidobacterium breve NCIMB 30441*, *Bifidobacterium longum NCIMB 30435*, *Bifidobacterium infantis NCIMB 30436*, *Lactobacillus acidophilus NCIMB 30442*, *Lactobacillus plantarum NCIMB 30437*, *Lactobacillus paracasei NCIMB 30439*, *Lactobacillus delbrueckii subsp* and *Bulgaricus NCIMB 30440.*	8 weeks	Patients with depression had significantly lower HAM-D scores and significantly lower activation of the left and right putamen.
2023 [173]	Adults with MDD aged 18 to 55 years taking antidepressants with an incomplete response (*n* = 49)	*Bacillus subtilis*, *Bifidobacterium bifidum*, *Bifidobacterium breve*, *Bifidobacterium infantis*, *Bifidobacterium longum*, *Lactobacillus acidophilus*, *Lactobacillus delbrueckii subsp bulgaricus*, *Lactobacillus casei*, *Lactobacillus plantarum*, *Lactobacillus rhamnosus*, *Lactobacillus helveticus*, *Lactobacillus salivarius*, *Lactococcus lactis*, and *Streptococcus thermophilus*	8 weeks	Probiotics can assist selective serotonin reuptake inhibitors to improve clinical outcomes in patients with depression.
2025 [147]	Male volunteers with a body mass index (BMI) greater than 25 and aged over 65 years (*n* = 67)	*Lactobacillus helveticus R0052* and *Bifidobacterium longum R0175*	8 weeks	Probiotics were able to significantly reduce geriatric depression Scale 15 scores in obese older adults and significantly improve total antioxidant capacity.

**Table 3 cimb-47-00487-t003:** Characteristics of the gut microbiota in populations exposed to acute and chronic high-altitude conditions.

Population	Key Microbiota Changes	Metabolic/Clinical Correlates
Acute exposure	- Increased aerobic bacteria (*Escherichia*, *Enterococcus*) - Reduced anaerobes (*Blautia*, *Bifidobacterium*) [18]	Gut barrier dysfunction, acute mountain sickness (AMS) [170]
Chronic adaptation	- Elevated *Firmicutes*, *Actinomyces*, *Clostridium*- Higher SCFA production (*Lachnospiraceae*, *Clostridiaceae*) [147,174]	Improved hypoxia tolerance, reduced inflammation

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
