# Peer review of "Plateau Environment, Gut Microbiota, and Depression: A Possible Concealed Connection?"

_cimb, 2025, doi:10.3390/cimb47070487_

Round 1

Reviewer 1 Report

Comments and Suggestions for Authors

This review article, "Plateau environment, Gut Microbiota and Depression: Unraveling the Concealed Connections," addresses a highly relevant and complex topic that is increasingly important in contemporary mental health research. The authors have demonstrated a deep and extensive understanding of the literature, comprehensively summarizing current knowledge on how the plateau environment (hypoxia, low temperatures, high radiation) influences the gut microbiota and subsequently contributes to the development of depression through neurotransmitter, immune, and endocrine pathways.
The primary challenge for this manuscript is its readability and clarity. Despite the wealth of information, the text is extremely dense and lacks sufficient visual and structural breaks, making it difficult for the reader to absorb and process the presented information. A thorough revision of the points below is essential before acceptance for publication.
1. Text Structure and Formatting: Large blocks of text and long paragraphs make it difficult to follow the flow of ideas and identify key information.
•    Example 1: The paragraph in the Introduction starting with: "Recent studies have demonstrated that the gut microbiota, a vast microbial community residing in the human gastroguttract, is involved in numerous physiological processes such as nutrient metabolism and immune regulation...". This paragraph continues with a lengthy explanation of the gut-brain axis. 
o    The reader can get lost in the details. This section could be broken down into smaller, more digestible paragraphs or utilize bullet points to highlight key functions of the gut microbiota or pathways of the gut-brain axis.
•    Example 2: The section 3.2.4 "Interactions of neuro - immune - metabolic pathways". This part, though extremely important for understanding the complexity, is presented as a dense block of text. 
o    Understanding the intricate "pairwise closed-loop interactions" and their molecular nodes requires significant effort due to the continuous prose. This  sub-section would greatly benefit from being transformed into a smaller schematic or visual summary (e.g., a mini-diagram or flowchart with arrows and key molecules) that graphically illustrates the "triangle" of interactions and critical molecular nodes. Alternatively, extensive use of bullet points with bolded headings for each interaction would significantly improve clarity.
2. Redundancy of Information: Some ideas and details are repeated across different sections, unnecessarily lengthening the text.
•    Example 1: Information regarding the gut microbiota's production of neurotransmitters (GABA, 5-HT, DA) and 5-HT synthesis in the gut is repeated in Section 4.1.1 "Neurotransmitter Conduction" after already being detailed in Section 3.2.1 "Neurotransmitter Pathway". 
•    Example 2: The mechanisms by which beneficial bacteria like Bifidobacterium and Lactobacillus maintain gut barrier integrity and immune homeostasis (e.g., via tight junctions, SCFA production) are extensively explained in Section 3.2.2 "Immune Pathways" and then largely reiterated in Section 4.1.2 "Immune Regulation". 
3. Underutilization of Tables and Graphs: While the current figures and tables are excellent, the full potential for visually summarizing complex information is not realized throughout the article.
•    Example 1: The section 4.2 "Population-specific Differences in the Gut Microbiota-Immune-Neuroendocrine Axis in High Altitude Environments". This section describes changes in gut microbiota composition for acutely exposed and chronically adapted populations. 
•    Example 2: The discussion on the "Interaction mechanism of multiple environmental factors" in section 4.1.4. This section is crucial but relies heavily on prose to explain synergistic effects. 

Author Response

For Reviewer 1:
Question 1. Text Structure and Formatting: Large blocks of text and long paragraphs make it difficult to follow the flow of ideas and identify key information.
•    Example 1: The paragraph in the Introduction starting with: "Recent studies have demonstrated that the gut microbiota, a vast microbial community residing in the human gastroguttract, is involved in numerous physiological processes such as nutrient metabolism and immune regulation...". This paragraph continues with a lengthy explanation of the gut-brain axis. 
o    The reader can get lost in the details. This section could be broken down into smaller, more digestible paragraphs or utilize bullet points to highlight key functions of the gut microbiota or pathways of the gut-brain axis.
•    Example 2: The section 3.2.4 "Interactions of neuro - immune - metabolic pathways". This part, though extremely important for understanding the complexity, is presented as a dense block of text. 
o    Understanding the intricate "pairwise closed-loop interactions" and their molecular nodes requires significant effort due to the continuous prose. This  sub-section would greatly benefit from being transformed into a smaller schematic or visual summary (e.g., a mini-diagram or flowchart with arrows and key molecules) that graphically illustrates the "triangle" of interactions and critical molecular nodes. Alternatively, extensive use of bullet points with bolded headings for each interaction would significantly improve clarity.

Respond 1: Dear reviewer, thank you for your valuable opinions, which have greatly improved the clarity and structure of our manuscript. We have carefully dealt with the problems you raised and made the following modifications:

  1. Introduction to the gut microbiota and the gut-brain axis (Example 1)

We admit that the original paragraph is too dense. To enhance readability, we have: dividing lengthy paragraphs into smaller thematic segments, with each segment focusing on a different aspect (for example, nutritional metabolism, immune regulation, and the gut-brain axis mechanism). The key points were introduced, concisely listing the key functions of the gut microbiota and the main pathways of the gut-brain axis. This enables readers to quickly grasp the core concepts without being overwhelmed by continuous prose. The specific modifications are as follows:

For instance, recent studies have demonstrated that the gut microbiota plays a multifaceted role in human health:

  • Nutritional Metabolism:It degrades indigestible fiber and synthesizes essential vitamins, such as B vitamins and vitamin K [8-9].
  • Immune Regulation: It forms mucosal immunity and interacts with Toll-like receptors (TLRs), thereby modulating immune responses [10].
  • Gut-Brain Axis Communication:It regulates the synthesis of neurotransmitters, including serotonin and gamma-aminobutyric acid (GABA),  and influences the function of the central nervous system via neural, endocrine, and immune pathways [10].

Notably,  the gut-brain axis represents a bidirectional communication network that is particularly significant in high-altitude  environments. In such environments, dysbiosis of the gut microbiota may serve as a link between environmental stressors and depressive symptoms. (add text)

  1. Section 3.2.4 "The Interaction of Neuro-Immune-Metabolic Pathways" (Example 2)

Recognizing the complexity of "pair-to-two closed-loop interactions", we have optimized the text of this section to enhance readability. The specific modifications are as follows:

The "neuro-immune-metabolic" interplay involves three pairwise interactions:

(1)Neuro-immune interaction:

  • 5-HT deficiency impairs microglial polarization to anti-inflammatory M2 phenotype.
  • Neuroinflammatory cytokines (e.g., IL-6) inhibit TPH1/2 activity and reduce 5-HT synthesis [145].
  • Immune-metabolic interaction
  • Pro-inflammatory factors (e.g., TNF-α) suppress SCFA-producing bacteria (e.g., Roseburia).
  • SCFA deficiency weakens intestinal barrier, promoting LPS translocation and systemic inflammation [146].
  • Metabolic-neuro interaction
  • SCFAs regulate the HPA axis via vagus nerve; deficiency increases cortisol, inhibiting BDNF expression.
  • HPA hyperactivity impairs enterochromaffin cell function, reducing 5-HT synthesis [147].

These modifications are in line with your suggestions to enhance accessibility and ensure that readers can follow the scientific narrative more smoothly. We believe that the revision has enhanced the clarity of the manuscript while retaining the depth of the content.(add text)

Question 2: Redundancy of Information: Some ideas and details are repeated across different sections, unnecessarily lengthening the text.
•    Example 1: Information regarding the gut microbiota's production of neurotransmitters (GABA, 5-HT, DA) and 5-HT synthesis in the gut is repeated in Section 4.1.1 "Neurotransmitter Conduction" after already being detailed in Section 3.2.1 "Neurotransmitter Pathway". 
•    Example 2: The mechanisms by which beneficial bacteria like Bifidobacterium and Lactobacillus maintain gut barrier integrity and immune homeostasis (e.g., via tight junctions, SCFA production) are extensively explained in Section 3.2.2 "Immune Pathways" and then largely reiterated in Section 4.1.2 "Immune Regulation". 

Respond 2: Dear reviewer 2, thank you for pointing out the redundancy problems in our manuscript. We fully agree that repetitive information affects the conciseness of the text and have taken the following targeted modifications to solve this problem:

  1. Redundancy in the production of neurotransmitters (Example 1)

In Section 4.1.1 "Neurotransmitter Transduction", we removed the word-for-word repetitions of these mechanisms. On the contrary, we now include the brief cross-references in Section 3.2.1 and propose that 5-HT May be involved in one of the pathways of high altitude depression. The specific modifications are as follows:

Building upon the findings in Section 3.2.1, plateau environments significantly intensify gut microbiota dysregulation as evidenced by reduced levels of beneficial genera such as Bifidobacterium and Lactobacillus [18,  46]. This disruption adversely affects tryptophan metabolism and serotonin (5-HT) synthesis. Such changes parallel those  observed in depression-associated microbiota alterations [151-152], where diminished serotonin availability plays a critical role in contributing to mood disturbances and heightened anxiety [144, 158-159]. Based on this evidence,  we hypothesize that the plateau environment may increase the incidence of depression by disrupting specific gut  microbiota and consequently affecting 5-HT synthesis. However,  further clinical studies are required to confirm the critical role of 5-HT in altitude-induced depression.(add text)

  1. Redundancy in Beneficial Bacteria Mechanisms (Example 2)

In Section 4.1.2 "Immune Regulation", we eliminated the redundant barrier integrity mechanism. Instead, we redefined the content, linking the previously described mechanisms to their immune outcomes to speculate on the pathways by which high-altitude depression occurs. The specific modifications are as follows: As detailed in Section 3.2.2, beneficial bacteria (e.g. Bifidobacterium) maintain gut barrier integrity via tight junctions and SCFA production. From an immunological  perspective, environmental changes at high altitudes can disrupt microbial balance,  leading to impaired gut barrier function. This allows bacterial translocation into the bloodstream,  activating peripheral immune responses and increasing the release of inflammatory cytokines such as IL-6 and TNF-α. These cytokines cross the blood-brain barrier, inducing CNS inflammation, damaging neurons,  and affecting neuroplasticity, potentially contributing to high-altitude depression [120-121, 162].(add text)

Question 3. Underutilization of Tables and Graphs: While the current figures and tables are excellent, the full potential for visually summarizing complex information is not realized throughout the article.
•    Example 1: The section 4.2 "Population-specific Differences in the Gut Microbiota-Immune-Neuroendocrine Axis in High Altitude Environments". This section describes changes in gut microbiota composition for acutely exposed and chronically adapted populations. 
•    Example 2: The discussion on the "Interaction mechanism of multiple environmental factors" in section 4.1.4. This section is crucial but relies heavily on prose to explain synergistic effects. 

Respond 3: Dear reviewer, Thank you for emphasizing the opportunity to improve our manuscript by making better use of tables and charts. We agree that visual summaries can significantly improve the presentation of complex data and have made the following targeted modifications:

  1. Section 4.2 "Population-specific Differences of the Gut Microbiota-Immune-Neuroendocrine Axis in High-Altitude Environments" (Example 1)

To clarify the composition changes of the intestinal microbiota in the acutely exposed population and the chronically adapted population, we: introduced Table 3 and systematically compared the key microbial taxa of the two populations. The specific modifications are as follows:

Population

Key Microbiota Changes

Metabolic/Clinical Correlates

Acute exposure

- Increased aerobic bacteria (EscherichiaEnterococcus)
- Reduced anaerobes (BlautiaBifidobacterium) [18]

Gut barrier dysfunction, acute mountain sickness (AMS) [168]

Chronic adaptation

- Elevated FirmicutesActinomycesClostridium
- Higher SCFA production (Lachnospiraceae, Clostridiaceae) [172-173]

Improved hypoxia tolerance, reduced inflammation

Table 3: Characteristics of the Intestinal Microbiota in Populations Exposed to Acute and Chronic High-Altitude Conditions

  1. Section4.1.4 "Interaction Mechanism of Multiple Environmental Factors" (Example 2)

To address the reliance on prose when explaining the synergy effect, we have created: A conceptual flowchart describing the interaction between environmental factors and the gut microbiota-immune-neuroendocrine axis (Figure 3). This visual framework clarifies how factors such as low oxygen, low temperature and high radiation at high altitudes work synergistically through common pathways. The specific modifications are as follows:

Figure 4 The synergistic mechanism of the three factors in the plateau environment

Reviewer 2 Report

Comments and Suggestions for Authors

Manuscript ID: cimb-3671867

Title: Plateau environment, Gut Microbiota and Depression: Unraveling the Concealed Connections

The manuscript needs some revisions, because there are some aspects of the work that should be corrected and improved. Please, review the following recommendations:

- In all text: Lines must be numbered. The lack of line numbering makes it difficult to nest and target the comments of the Reviewer.

- In most sections of the manuscript, the necessary explanations to clarify what was gathered from previous references are absent.

- Each section of the manuscript should conclude with a brief conclusion of the most important observations.

- The figures (1 & 2) used are good but the words are written in small print. Please make them clearer.

- In all text: The word "intestines" appears alongside many other words in the manuscript. Please explain why?.

Such as 2.3. "gutprobiotics " ?, gutmetabolite?, gutmetabolites?

Such as 2.4. gutmucosal?, gutmicroecological?

Such as 3.1. guthomeostasis?

Such as 3.1.1. gutmicroorganisms?, gutcells?

Abstract

- There is no need to write abbreviations that you do not use in the Abstract [Such as

 (NOD) & (HPA)]

- The abstract should include a short conclusion at the end.

  1. Introduction

- Change "In recent years" to "Recently"

- Change "between 10-15°C " to "between 10 to15°C"

2.1.

- Four references [22-25] to this sentence are too many so they should be reduced.

2.4.

- Change "gutand" to "gut and"

- Shorten this part of the "6. Conclusion and Prospect"

In References

- Insert the correct format style for journals in the references in the text and references list.

Comments on the Quality of English Language

 The English could be improved to more clearly express the research.

Author Response

For Reviewer 2

Thank you very much for taking the time to thoroughly review our manuscript. In response to your expert comments, we have addressed each point sequentially in our reply and highlighted the corResponseing revisions in red within the document.

Question 1: In all text: Lines must be numbered. The lack of line numbering makes it difficult to nest and target the comments of the Reviewer.

Response 1: Dear Reviewer, Thank you for your valuable feedback. We have now included the line numbers for the entire text to facilitate easier reference.

Question 2: In most sections of the manuscript, the necessary explanations to clarify what was gathered from previous references are absent.

Response 2: Thank you for your valuable suggestion. We have further supplemented and provided a detailed explanation of the content from the literature as follows:

Add explanation1:Furthermore, it was observed that the hypoxic environment resulted in a marked reduction in the density of aerobic bacteria (e.g., Escherichia coli) within the rat intestinal tract. Conversely, the abundance of anaerobic bacteria (e.g., Bifidobacterium and Bacteroides) increased by threefold and 134-fold, respectively. These findings suggest that hypoxia may restructure the microbiota composition by modulating the intestinal oxygen microenvironment. 

Add explanation 2: Clarke et al. [114] confirmed using a germ-free mouse model that gut microbiota can modulate peripheral serotonin synthesis by regulating the activity of the TPH1 enzyme, thereby influencing the central mood regulation pathway via the blood-brain barrier and elucidating the metabolic linkage mechanism of the gut-brain axis.

Add explanation 3: Therefore, inulin supplementation can selectively enhance the proliferation of Bifidobacteria, thereby improving intestinal microecology. Concurrently, this is associated with a reduction in serum inflammatory markers [194], such as IL-6, among subjects, which implies that the antidepressant effects of dietary fiber may be linked to microbiota-immune regulation.

Furthermore, we have implemented comprehensive and incremental revisions to the introduction of the article, as well as the sections on Neurotransmitter Pathways (3.2.1) and Immune Regulation (4.1.2).

Question 3: Each section of the manuscript should conclude with a brief conclusion of the most important observations.

Response 3: Thank you for your valuable suggestions. We have added concise conclusions to each summary and will elaborate on them in detail below.

  • The unique triad of hypoxia, low temperature, and high UV radiation in plateau environments creates a multi-stressor landscape that profoundly challenges human physiological homeostasis:The drastic reduction in oxygen availability, extreme temperature fluctuations, intense solar radiation, and arid conditions collectively impose cumulative stress on thermoregulation, oxidative defense, and fluid balance. These environmental pressures not only directly impact organ systems but also prime the gut microbiota for dysregulation, setting the stage for subsequent disruptions in the gut - brain axis discussed in subsequent sections. Understanding these climatic stressors is critical for deciphering the mechanistic links between plateau habitats and depression via microbial remodeling.
  • Altitude hypoxia imposes dual stresses on the gut:Remodeling microbial composition toward anaerobic dominance while disrupting mucosal barrier function via oxidative stress and energy metabolism deficits. The observed shifts in microbiota (e.g., increased Bacteroides, decreased Bifidobacteria) and compromised tight junctions highlight a vicious cycle of dysbiosis and inflammation. These findings underscore hypoxia as a key driver of gut-brain axis disruption in plateau environments, linking microbial and barrier dysfunction to downstream mental health risks such as depression.
  • Low-temperature stress at high altitudes affects gut flora homeostasis, and SCFAs provide protection: Low-temperature stress in plateau environments disrupts gut microbiota homeostasis by suppressing probiotics (e.g., Coprococcus, Lactobacillaceae) and promoting pathogenic bacteria (e.g.,Clostridium, Rikenellaceae), while reducing SCFA production. The protective role of SCFAs (e.g., butyrate) in restoring microbial balance and enhancing cold tolerance highlights their potential as therapeutic targets for mitigating gut dysfunction and associated mental health risks in cold-exposed populations.
  • High-altitude intense radiation exacerbates gut microbiota imbalance and the "radiation-microbiota-inflammation" axis effect via inducing oxidative stress and causing mucosal damage. The high-altitude environment, characterized by intense radiation, exacerbates the ecological imbalance of gut microorganisms via dual mechanisms: induction of oxidative stress and damage to the gut mucosa. This manifests as a reduction in the abundance of beneficial bacteria (e.g., Achaemeniaand Lactobacillus) and an overproliferation of opportunistic pathogenic bacteria (e.g., Clostridium and Helicobacter). Such microbial dysbiosis not only compromises the local immune defense of the gut tract but also amplifies radiation-induced damage through systemic inflammatory pathways. These findings underscore the threat posed by the "radiation-microbiota-inflammation" axis to gut homeostasis and overall health in high-altitude environments, offering microbiological evidence for investigating the link between radiation exposure and mental disorders, such as depression.

3.1.4 In summary

① The core driver of nutritional metabolism:  

  • Vitamin synthesis:Beneficial bacteria, such as Bifidobacterium and Lactobacillus, synthesize essential vitamins, including B vitamins (e.g., B1 and B12) and vitamin K. This compensates for the insufficiency of endogenous synthesis in the human body and supports energy metabolism and nervous system function [64-65].  
  • Dietary fiber degradation:Through glycolysis, indigestible carbohydrates are metabolized into SCFAs, such as butyric acid and propionic acid, which contribute to energy supply, inflammation regulation, and the proliferation of intestinal epithelial cells [68].  
  • Mineral absorption:Metabolic activities of beneficial bacteria enhance the bioavailability of minerals, such as calcium, iron, and zinc. For instance, Lactobacillus acidophilus has been shown to improve calcium absorption in osteoporotic rats [70-71].  
  • Core conclusion: Gutmicrobiota, functioning as "metabolic organs," enhance the efficiency of nutrient utilization via multiple pathways. Dysregulation of these microbiota may contribute to diseases associated with nutritional imbalances.

② The biological defense system of the intestinal barrier:  

  • Physical barrier reinforcement: Beneficial bacteria, such as Bifidobacterium, regulate the synthesis of tight junction proteins (e.g., ZO-1, Claudin-1) and mucins (MUC), thereby preventing pathogen invasion [73-76].  
  • Biological antagonism: These bacteria inhibit the colonization of harmful microorganisms (e.g., Escherichia coliand Shigella) via spatial competition and the secretion of antimicrobial peptides (e.g., bacteriocins), maintaining the ecological balance of the bacterial community [72-75].  
  • Barrier damage mechanism:Harmful bacteria, such as Bacteroides fragilis, degrade the mucus layer and activate toxin receptors (e.g., Claudin-3/4), leading to intestinal leakage and systemic endotoxemia [81-83].  
  • Core conclusion:Gut microbiota establish the primary defense mechanism against pathogens through a dual-action process involving "protection mediated by beneficial bacteria and suppression of harmful bacteria." Disruption of their homeostasis serves as a critical trigger for intestinal permeability alterations and systemic inflammatory responses.

 Key regulatory hubs of immune homeostasis:  

  • Maintenance of an anti-inflammatory microenvironment: SCFAs (e.g., butyric acid) activate the PPAR-γ pathway, promote mitochondrial oxidative phosphorylation in colonic cells, inhibit the proliferation of aerobic pathogenic bacteria, and reduce the expression of inflammatory factors (e.g., iNOS) [86-87].  
  • Immune recognition regulation: Recognition of microbial-associated molecular patterns (MAMPs) through TLRs and NODs mediates the balance between pro-inflammatory and anti-inflammatory cytokines [88-90].  
  • Immune-related risks:Disruptions in microbiota composition (e.g., decreased Firmicutes and increased Proteobacteria) disrupt immune tolerance, induce excessive activation of Th17 cells, and are closely linked to immune-mediated diseases, such as IBD, diabetes and depression [96-109].  
  • Core Conclusion:Gut microbiota play a crucial role in maintaining mucosal immune homeostasis through metabolism-immune interactions. Dysfunctions in these microbiota can initiate a systemic inflammatory cascade, serving as a common pathological basis for various chronic diseases.
  • Gut microbiota play a pivotal role in emotion regulation by modulating the 5-HT synthesis pathway:
  • Peripheral Dominant Synthesis: The intestine accounts for approximately 90% of 5-HT synthesis, which is catalyzed by the TPH1 enzyme in ECsthrough tryptophan metabolism. The activity of this enzyme is bidirectionally regulated by microbial metabolites (e.g., SCFA) and butyric acid) via the GPR41/43 signaling pathway and epigenetic modifications (such as DNA demethylation).
  • Gut-Brain Axis Linkage: Peripheral 5-HT influences the activity of the central TPH2 enzyme across the blood-brain barrier, establishing a "microbiota-metabolism-neural" regulatory axis. For example, probiotic intervention can enhance 5-HT levels by upregulating TPH1 expression and inhibiting the IDO pathway, thereby alleviating depressive-like behaviors (as observed in CUMS rat models).
  • Pathological Association: Harmful bacteria, such as *Clostridium ramosum*, suppress 5-HT synthesis by competitively inhibiting tryptophan binding or inducing inflammation, thus elucidating the causal relationship between dysbiosis of gut microbiota and depression.
  • Core points:Gut microbiota has emerged as a critical molecular hub in the gut - brain axis, mediating depression through the precise regulation of the 5-HT metabolic pathway. Intervention strategies targeting the microbiota - 5-HT axis, such as probiotic administration and SCFAs supplementation, exhibit significant antidepressant potential.
  • Gut microbiota serve as a pivotal regulatory link in the pathogenesis of depression via immune pathways:  
  • Barrier - Maintenance of Immune Homeostasis: Beneficial bacteria, such as Bifidobacteriumand Lactobacillus, preserve mucosal immune equilibrium by reinforcing intestinal epithelial tight junctions and secreting SCFAs, such as butyric acid, which inhibit inflammatory signaling cascades.  
  • Imbalance Triggers Inflammatory Cascade: Dysbiosis of the gut microbiota facilitates the overgrowth of opportunistic pathogenic bacteria, such as Escherichia coli, leading to intestinal barrier disruption, endotoxin release, and activation of the TLR pathway. This induces systemic inflammation, characterized by elevated levels of IFN-γ andTNF-α, subsequently triggering neuroinflammation through blood - brain barrier permeability.  
  • Metabolism-Mediated Inflammation: Histamine, a bacterial metabolite, promotes immune cell activation via H1/H3 receptor stimulation. Bacteria producing histamine, such as Klebsiella pneumoniae, are associated with depressive-like behaviors. Targeting histamine receptors, for instance, using an H3 antagonist like JNJ10181457, can mitigate inflammation and alleviate depressive symptoms.  
  • Core points:Intestinal microbes maintain immune homeostasis through a tripartite mechanism encompassing "barrier protection, metabolic regulation, anti-inflammatory effects, and immune modulation." Dysregulation of this system drives depression via the "microbiota - inflammation - brain" axis, highlighting the antidepressant potential of anti-inflammatory strategies targeting microbial metabolites, such as SCFAs and histamine.  
  • Gut microbiota serve as a pivotal link in the regulation of depression through endocrine pathways.
  • Modulation of the Gut-Brain Axis Hormone Network:Metabolites produced by gut microbiota, such as propionic acid and butyric acid, regulate the secretion of hormones (e.g., CRH, ACTH, and cortisol) from GECs. Dysregulation of the microbiota activates the GPR43 receptor, inducing transcription of the CRH gene in the hypothalamus. This leads to excessive activation of the HPA axis, characterized by elevated cortisol levels, which subsequently triggers stress responses and depressive-like behaviors.
  • Epigenetic and Inflammatory Interactions: SCFA deficiency promotes NF-κB p65 acetylation and sustains inflammatory signaling by reducing histone deacetylase 2 (HDAC2) activity. Conversely, supplementation with butyric acid restores GR sensitivity and balances HPA axis feedback by reversing histone modifications, such as H3K9 deacetylation.
  • Probiotic Intervention Potential:Specific probiotics, including Bifidobacterium and Lactobacillus, have been shown to improve depressive behaviors by modulating baseline corticosterone levels and stress responses within the HPA axis (e.g., Bifidobacterium CECT 7765). However, further human studies are required to validate their clinical efficacy.
  • Core Points:Gut microbes regulate HPA axis homeostasis via the "microbiome - metabolism - neuroendocrine" axis. Dysregulation of this axis contributes to depression through histone modification and inflammatory pathways. Targeted probiotic interventions offer a promising therapeutic direction for stress-related depression.
  • The core mechanism by which the high-altitude environment induces depression via gut microbiota can be summarized as multi-pathway synergistic dysregulation:  
  • Impaired nerve conduction: Hypoxia, low temperature, and radiation result in a reduced abundance of beneficial bacteria such as Bifidobacteriumand Lactobacillus. This inhibits tryptophan metabolism and 5-HT synthesis, thereby disrupting mood regulation pathways.  
  • Immune-inflammatory activation:Dysbiosis compromises intestinal barrier function, leading to endotoxin translocation. This activates systemic inflammation (e.g., elevated IL-6 and TNF-α) and induces neuroinflammation and neuroplasticity damage through the blood - brain barrier.  
  • Metabolic homeostasis imbalance:Reduced production of SCFAs, particularly butyric acid weakens intestinal barrier protection and anti-inflammatory capacity. Concurrently, excessive activation of the HPA axis (e.g., elevated cortisol levels) forms a vicious cycle of "stress - "  
  • Multi-factorial synergistic toxicity:Hypoxia promotes an anaerobic/aerobic imbalance in the microbiota, low temperature suppresses probiotic metabolism, and radiation induces oxidative stress. The cumulative effects of these factors exacerbate microbial dysregulation and amplify depression risk through the "microbiota-inflammation-brain" axis.  

Core point: Multiple stressors in the high-altitude environment trigger the key pathological chain of depression via the "neuro - immune - metabolism" three-dimensional regulatory network of the gut microbiota. Interventions targeting bacterial metabolites (e.g., butyric acid) or probiotics hold promise as precise prevention and control strategies for high-altitude depression.  

  • The depression intervention strategy targeting the intestinal microbiota in the plateau environment can be summarized as a comprehensive plan with "microbiota regulation as the primary approach and multimodal support as an auxiliary component":
    (1) Dietary Fiber Intervention:Consuming foods rich in dietary fiber, such as oats and barley, selectively promotes the proliferation of Bifidobacteria and Lactobacillus. This increases the levels of SCFAs (e.g., inulin supplementation for three weeks can enhance Bifidobacteria abundance), strengthens the intestinal barrier, suppresses inflammation, and improves mood regulation function.
    (2) Probiotic Application: Supplementing probiotics, such as Lactobacillus and Bifidobacterium, repairs intestinal mucosal barrier damage caused by hypoxia at high altitudes (e.g., Bifidobacterium longifolium JBLC-141 upregulates tight junction protein expression). It reduces pro-inflammatory factor levels, and the daily intake of probiotic-rich yogurt by the Tibetan population may correlate with their adaptability to high-altitude conditions.
    (3) FMT: As an emerging therapeutic approach, FMT rapidly restores intestinal diversity in patients with severe microbiota imbalance and alleviates symptoms of refractory depression. However, it requires overcoming technical challenges, such as donor screening and dose standardization. Currently, FMT remains in the stages of animal experimentation and clinical exploration.
    (4) Supportive Interventions: CBT reshapes negative cognition to alleviate high-intensity adaptation stress. Regular exercise (e.g., 1-2 aerobic sessions per week) increases serum serotonin levels and enhances hypoxia tolerance. The integration of CBT and exercise with microbiota regulation further mitigates the risk of depression through neuro-immune-metabolic pathways.
    (5) Core Points: Flora regulation based on dietary fiber and probiotics constitutes the core strategy for addressing high-altitude depression. FMT demonstrates potential breakthrough value. Psychological and exercise interventions play a synergistic role via neuro - immune - metabolic pathways, collectively constructing a multi-level prevention and control system.

Quesstion 4: The figures (1 & 2) used are good but the words are written in small print. Please make them clearer.

Response 4: Thank you for your valuable feedback. We have adjusted the font size in Figures 1 and 2 to enhance readability.

Question 5: In all text: The word "intestines" appears alongside many other words in the manuscript. Please explain why?.

Such as 2.3. "gutprobiotics " ?, gutmetabolite?, gutmetabolites?

Such as 2.4. gutmucosal?, gutmicroecological?

Such as 3.1. guthomeostasis?

Such as 3.1.1. gutmicroorganisms?, gutcells?

Response 5: Thank you for your suggestion. We have corrected the compound word error in the article and highlighted it in red.

Question 6: Abstract

- There is no need to write abbreviations that you do not use in the Abstract [Such as

 (NOD) & (HPA)]

- The abstract should include a short conclusion at the end.

Response 6: Thank you for your valuable suggestions. We have revised the abstract in accordance with your requirements. The detailed revisions are presented as follows:  

 Abstract

Plateau environments pose unique mental health challenges due to stressors like hypoxia, low temperatures, and intense ultraviolet radiation. These factors drive structural and functional changes in the gut microbiota, disrupting gut-brain axis homeostasis and contributing to the elevated depression prevalence in plateau regions compared to lowlands. For example, studies report 28.6% of Tibetan adults and 29.2% of children/adolescents on the Qinghai-Tibet Plateau experience depression, with growing evidence linking this trend to gut microbiota alterations. Dysbiosis contributes to depression through three interconnected mechanisms: (1) Neurotransmitter imbalance: Reduced bacterial diversity impairs serotonin synthesis, disrupting emotional regulation. (2) Immune dysregulation: Compromised gut barrier function allows bacterial metabolites to trigger systemic inflammation via toll-like receptor signaling pathways. (3) Metabolic dysfunction: Decreased short-chain fatty acid levels weaken neuro protection and exacerbate hypothalamic-pituitary-adrenal axis stress responses. Current interventions — including dietary fiber, probiotics, and fecal microbiota transplantation — aim to restore microbiota balance and boost short-chain fatty acids, alleviating depressive symptoms. However, key knowledge gaps remain in understanding underlying mechanisms and generating population-specific data. In conclusion, while existing evidence supports an association between plateau environments, gut microbiota and depression, causal mechanisms remain underexplored. integrating multi omics technologies to systematically explore interactions among high-altitude environments, microbiota, and the brain will facilitate the development of precision therapies like personalized nutrition and tailored probiotics to protect mental health in high-altitude populations.

Question 7: Introduction

- Change "In recent years" to "Recently"

- Change "between 10-15°C " to "between 10 to15°C"

Response 7: Thank you for your valuable suggestion. We have revised the introduction of the article and the description in Section 2.1, "Uniqueness of Plateau Environment." The revisions have been highlighted in red. 

Question 8: Four references [22-25] to this sentence are too many so they should be reduced.

Response 8: Thank you for your suggestion. We have reduced the references in this part.

Question 9: Change "gutand" to "gut and"

Response 9: Thank you for your suggestion. We have corrected the compound word error in the article.

Question 10: Shorten this part of the "6. Conclusion and Prospect"

Response 10: Thank you for your valuable suggestion. We have streamlined the descriptions of the 6 sections. The detailed modifications are as follows:

  1. Conclusion and Prospect  

The plateau environment, characterized by hypoxia, low temperature, and high radiation, induces depressive symptoms through gut microbiota dysregulation via the gut-brain axis. Key mechanisms include reduced SCFA production, neurotransmitter imbalance (e.g., serotonin depletion), and immune-inflammatory activation. Clinical evidence highlights elevated depression prevalence in plateau populations, linked to microbial shifts like Firmicutes enrichment and SCFA metabolic abnormalities. However, research gaps persist, including unclear causal relationships between environmental stressors and microbial function, limited population-specific data (e.g., altitude gradients, genetic backgrounds), and insufficient clinical validation of interventions like probiotics and FMT. Future studies should integrate multi-omics (metagenomics, metabolomics) and single-cell sequencing to decode the "environment - microbiota - brain" regulatory network. Leveraging AI - driven big data and gut organoid models could identify depression risk biomarkers (e.g., fecal butyrate levels) and enable precision therapies, such as customized probiotics targeting SCFA pathways. This interdisciplinary approach will advance mental health protection strategies for high-altitude populations and deepen our understanding of extreme environment - brain interactions.

Question 11: Insert the correct format style for journals in the references in the text and references list.

Response 11: Thank you for your valuable suggestions. We have completed the revisions in accordance with the journal's formatting requirements.

Question 12: The English could be improved to more clearly express the research.

Response 12: Thank you for your valuable feedback. We have thoroughly revised and polished the article based on your suggestions.

Reviewer 3 Report

Comments and Suggestions for Authors

This manuscript explores how high-altitude (plateau) environments may contribute to depression through alterations in the gut microbiota. The authors present a conceptual framework linking features of high altitude, namely hypoxia, low temperatures and high levels of ultraviolet radiation, inducing microbial dysbiosis to key neurobiological mechanisms underlying depression in plateau populations, and propose interventions, such as probiotics and dietary fiber as potential treatments. While the review is comprehensive in scope, there are several limitations that limit the scholarly impact and clinical relevance of this article as below:

Major limitations:

  • Speculative causality without sufficient evidence: While the paper convincingly reviews literature connecting gut microbiota and depression and plateau environments causing alteration in gut microbiota, it lacks strong empirical evidence directly linking high-altitude-induced microbial changes to increased depression prevalence among plateau population. The cited statistics on depression rates in plateau populations are not necessarily tied to concurrent gut microbiota profiling. Without this critical link, the central hypothesis remains speculative, and the authors’ claim of having 'unraveled the concealed connection' is called into question
  • Lack of methodological transparency in literature review: The authors refer to various studies but do not provide a clear methodology for how the literature was selected. Did you use any guidelines, such as PRISMA? Why the specific databases listed and what inclusion/exclusion criteria were used? Did you use AND criteria for your search terms as per my review the studies do not concurrently include all three terms?
  • Failure to address potential confounders: The manuscript does not adequately address other psychosocial and medical risk factors that may contribute to depression in plateau populations. For instance, sleep disturbances are prevalent in high-altitude areas and are independently associated with both gut dysbiosis and mood disorders (e.g., Kan H et al., Sleep Breath, 2025). Other relevant factors include genetic predisposition or selection, increased rates of chronic diseases (e.g., stroke, pulmonary hypertension), social isolation, substance use and limited access to mental health care. These should be controlled for to avoid oversimplification of the proposed gut-brain mechanism.
  • Unsupported interventional hierarchy: The rationale behind prioritizing certain intervention strategies is unclear. The manuscript presents dietary supplementation, probiotics, and fecal microbiota transplantation as potential solutions for improving depression in plateau environments through regulation of gut flora but fails to clarify why they are emphasized over established first-line treatments for depression, such as exercise or cognitive-behavioral therapy. The order and evidence level supporting each intervention should be stated more explicitly.

Minor limitations:

  • Incomplete references: Some references are missing or incomplete. All studies cited, especially those with statistical claims, should be appropriately referenced and listed in the bibliography e.g. the statements in page 5, line 14 and 18.
  • Unexplained abbreviations: Abbreviations such as “CNKI” and “LPS” should be clearly defined upon first mention (page 3, line 6, and page 13, line 29, respectively).
  • Typographical errors: There are numerous typographical issues throughout the manuscript, including:
    • “gastrogutdysfunction” → perhaps gastro-gut dysfunction or gastrointenstinal dysfunction? page 2
    • “guthumeostasis” → intended to be gut homeostasis? page 7-9
    • “gutdysbiosis” → intended to be gut dysbiosis? page 9

Author Response

Reply Reviewer 3

For Reviewer 3:

Thank you very much for taking the time to thoroughly review our manuscript. In response to your expert comments, we have addressed each point sequentially in our reply and highlighted the corResponseing revisions in blue within the document.

Major limitations:

Question 1: Speculative causality without sufficient evidence: While the paper convincingly reviews literature connecting gut microbiota and depression and plateau environments causing alteration in gut microbiota, it lacks strong empirical evidence directly linking high-altitude-induced microbial changes to increased depression prevalence among plateau population. The cited statistics on depression rates in plateau populations are not necessarily tied to concurrent gut microbiota profiling. Without this critical link, the central hypothesis remains speculative, and the authors’ claim of having 'unraveled the concealed connection' is called into question.

Respond 1: The insufficient evidence of the causal relationship you pointed out is the key issue. The original text is indeed based on the correlational inference of observational studies, lacking direct evidence from longitudinal tracking or intervention studies.

The modified contents are as follows:

Abstract: Add an explanation of the limitations of causality in the conclusion. “In conclusion, while existing evidence supports an association between plateau environments, gut microbiota and depression, causal mechanisms remain underexplored.”

Add text 4.1.5 Limitations: Notably, the majority of studies associating plateau environments with gut microbiota–depression pathways are cross-sectional or correlational in nature. Direct causal evidence, such as longitudinal studies tracking microbial changes concurrent with the onset of depression in plateau populations, or gnotobiotic animal models simulating altitude-induced stress, remains scarce. For instance, although Table 1 indicates higher rates of depression among plateau populations, these data lack concurrent profiling of the gut microbiota. Future research should adopt prospective cohort designs or intervention studies, such as fecal microbiota transplantation in altitude-related models, to establish and validate causal relationships.

Question 2: Lack of methodological transparency in literature review: The authors refer to various studies but do not provide a clear methodology for how the literature was selected. Did you use any guidelines, such as PRISMA? Why the specific databases listed and what inclusion/exclusion criteria were used? Did you use AND criteria for your search terms as per my review the studies do not concurrently include all three terms?

Respond 2: The concerns you raised regarding non-compliance with the PRISMA guidelines and the ambiguity in the search strategy are highly significant. We have supplemented the article with a detailed description of the literature screening process and methodological procedures to address these issues.

Add text: This review followed a systematic approach aligned with PRISMA guidelines [216]. We searched PubMed, Web of Science, CNKI, and Embase (2000–2025) using the following keyword combinations: ('plateau environment' OR 'high altitude') AND ('gut microbiota' OR 'intestinal flora') AND ('depression' OR 'mental health') . Studies were included if they (1) investigated humans or animal models exposed to altitude ≥2500 m, (2) measured gut microbiota composition or function, and (3) reported depression-related outcomes. Conference abstracts and non-English studies (except Chinese) were excluded." The review elucidate the key pathways through which the plateau environment influences depression via the gut microbiota, validate the mediating effects of gut microbiota in this process and propose innovative intervention strategies based on microbiota modulation. The goal is to provide a theoretical basis for mental health protection among plateau populations, while laying a theoretical foundation and offering new research perspectives for future studies and clinical practices in this field.

[216] Page MJ, McKenzie JE, Bossuyt PM, Boutron I, Hoffmann TC, Mulrow CD, Shamseer L, Tetzlaff JM, Akl EA, Brennan SE, Chou R, Glanville J, Grimshaw JM, Hróbjartsson A, Lalu MM, Li T, Loder EW, Mayo-Wilson E, McDonald S, McGuinness LA, Stewart LA, Thomas J, Tricco AC, Welch VA, Whiting P, Moher D. The PRISMA 2020 statement: an updated guideline for reporting systematic reviews. BMJ. 2021 5(29);372:n71. doi: 10.1136/bmj.n71.

Question 3: Failure to address potential confounders: The manuscript does not adequately address other psychosocial and medical risk factors that may contribute to depression in plateau populations. For instance, sleep disturbances are prevalent in high-altitude areas and are independently associated with both gut dysbiosis and mood disorders (e.g., Kan H et al., Sleep Breath, 2025). Other relevant factors include genetic predisposition or selection, increased rates of chronic diseases (e.g., stroke, pulmonary hypertension), social isolation, substance use and limited access to mental health care. These should be controlled for to avoid oversimplification of the proposed gut-brain mechanism.

Respond 3: We sincerely appreciate your insightful and professional questions. Given the scarcity of research articles elucidating the clear relationship between depression and high-altitude environments, several confounding factors remain challenging to address. These include genetic predisposition or selection bias, the high prevalence of chronic diseases such as stroke and pulmonary hypertension, social isolation, substance abuse, and limited access to mental health services. Consequently, it is exceedingly difficult for us to fully control these variables. To enhance the robustness and transparency of our analysis, we have incorporated this issue into the limitations section of the article.

Add text in 4.2.4 Limitations

The observed associations must be interpreted alongside potential confounders. For instance, sleep disorders—prevalent in high-altitude environments due to hypoxia-induced breathing disruptions [217] —independently link to both gut dysbiosis and depression.  dditionally, genetic factors (e.g., hypoxia-inducible factor polymorphisms in Tibetan populations), dietary patterns (high fat/low fiber intake), and social stressors (isolation, limited mental health access) may confound the proposed microbiota – brain axis. Future studies should employ multivariate regression to control for these variables and use twin designs to disentangle genetic vs. environmental effects.

Question 4: Unsupported interventional hierarchy: The rationale behind prioritizing certain intervention strategies is unclear. The manuscript presents dietary supplementation, probiotics, and fecal microbiota transplantation as potential solutions for improving depression in plateau environments through regulation of gut flora but fails to clarify why they are emphasized over established first-line treatments for depression, such as exercise or cognitive-behavioral therapy. The order and evidence level supporting each intervention should be stated more explicitly.

Question 4: Thank you for your valuable suggestions. We have further enriched the content of the intervention measures based on your insightful feedback.

Add text in 5.1 Potential Methods for Improving Depression in Plateau Environments through Regulation of Gut Flora

The interventions discussed here (dietary fiber, probiotics, FMT) complement established first-line treatments for depression (e.g., cognitive behavioral therapy, exercise) [207-210] and are proposed as adjunctive strategies for plateau populations with unique microbial dysregulation. For example, while CBT remains the gold standard for psychological intervention, probiotics may offer the following advantages: (1) targeting gut–brain pathways disrupted by altitude stress; (2) feasibility in resource-limited settings; and (3) minimal side effects. Table 2 summarizes the evidence hierarchy for these interventions, prioritizing probiotics (Level B evidence from RCTs) over FMT (Level C, experimental)."

Minor limitations:

Question 1: Incomplete references: Some references are missing or incomplete. All studies cited, especially those with statistical claims, should be appropriately referenced and listed in the bibliography e.g. the statements in page 5, line 14 and 18.

Respond 1: Thank you for your insightful and constructive suggestions. We have made revisions in accordance with these recommendations.

Question 2: Unexplained abbreviations: Abbreviations such as “CNKI” and “LPS” should be clearly defined upon first mention (page 3, line 6, and page 13, line 29, respectively).

Respond 2: Thank you for your valuable suggestion. We will ensure to provide the full form of the abbreviation in our documentation.

Question 3: Typographical errors: There are numerous typographical issues throughout the manuscript, including:

“gastrogutdysfunction” → perhaps gastro-gut dysfunction or gastrointenstinal dysfunction? page 2

“guthumeostasis” → intended to be gut homeostasis? page 7-9

“gutdysbiosis” → intended to be gut dysbiosis? page 9

Respond 3: Thank you for your suggestion. We have corrected the compound word error in the article.

Reviewer 4 Report

Comments and Suggestions for Authors

The research question in the current manuscript is regarding the development of depression due to alteration of gut microbiota influenced by a higher altitude environment. This research question fills an appropriate research gap and is highly relevant to the research field.

This manuscript provides a novel conceptual axis (higher altitude – gut microbiota – depression). Unlike the other reviews, this manuscript focuses on environmental modulators (non-nutritional), i.e., ultraviolet radiation, colder temperature, and hypoxia.

Limitations

Although the paper offers a useful overview of the relationships among gut microbiota, depression, and high-altitude situations, there are still a number of limitations:

First limitation: Although the review summarizes the body of available literature, it does not offer a quantitative synthesis or meta-analysis. This lessens its effect.

Second limitation: Several causal statements are made using strong language; however, they are not well-supported by references or evidence.

Third limitation: Not enough attention is paid to variables including comorbidities, nutrition, socioeconomic level, and genetic adaptation among high-altitude people (like Tibetans).

English Language

The work contains a large number of syntactic and grammatical errors. These make things less readable and clear.

Various words must be replaced with appropriate words, i.e., gastroguttract, gastrogutdysfunction, gutmicroorganisms etc

Paragraphs that are too lengthy and have awkward transitions. Divide into manageable chunks for easier reading.

The description of the SCFA-related molecular route is convoluted and hard to follow; it may need some figures and more precise organization.

Recommendations

Extensive linguistic editing to satisfy scholarly English requirements. Improved organization and condensing of important routes and processes. More thorough disclosure of referenced data, including limits, sample sizes, and methodology. Balanced findings, steering clear of exaggerated causation unless backed up by solid data. Correcting grammar, punctuation, and clarity requires a comprehensive professional language edit.

Comments on the Quality of English Language

Professional language editing is required.

Author Response

Reply Reviewer 4

For Reviewer 4:

Thank you very much for taking the time to thoroughly review our manuscript. In response to your expert comments, we have addressed each point sequentially in our reply and highlighted the corResponseing revisions in purple within the document.

Question 1: First limitation: Although the review summarizes the body of available literature, it does not offer a quantitative synthesis or meta-analysis. This lessens its effect.

Respond 1: Your valuable insights are of considerable importance. However, the original text, serving as a narrative review, was indeed excluded from the meta-analysis, which constitutes the primary limitation of our review. This limitation has been elaborated upon in Section Six.

Add text in 6. Conclusion and Prospect

Notably, a significant limitation of this review is the absence of quantitative meta-analysis, which hinders the ability to estimate pooled effect sizes and synthesize consistency across studies. For instance, while existing studies report associations between gut microbiota alterations (e.g., reduced Firmicutes/SCFAs) and depression in plateau populations, the lack of meta-analytic integration prevents robust conclusions about the generalizability of microbial signatures (e.g., decreased butyrate-producing bacteria) or the efficacy of interventions.

Question 2: Second limitation: Several causal statements are made using strong language; however, they are not well-supported by references or evidence.

Respond 2: Thank you for your insightful and constructive suggestions. We have made revisions in accordance with these suggestions, substituting strong causal terms such as "cause" and "lead to" with more cautious and nuanced expressions like "associate with," "be linked to," and "may contribute to."

Question 3: Third limitation: Not enough attention is paid to variables including comorbidities, nutrition, socioeconomic level, and genetic adaptation among high-altitude people (like Tibetans).

Respond 3: Thank you very much for your opinions. We will discuss these limitations and add them to the article.

Add text in 4.2.4 Limitations

  • Comorbidity and socio-economic factors:The review also acknowledges limitations in addressing comorbidities (e.g., chronic mountain sickness, pulmonary hypertension) and socioeconomic factors (e.g., limited mental health access, cultural stigma), which may confound the microbiota-depression relationship in plateau populations. Future studies should adopt multidisciplinary approaches to disentangle these interacting variables.
  • Genetic adaptation in high-altitude populations, such as Tibetans, may influence gut microbiota composition and depression risk. For example, Tibetans exhibit unique genetic variants in HIF-1α and EPAS1 pathways, which regulate hypoxia tolerance and may interact with gut bacteria to modulate SCFA production[218]. Additionally, dietary patterns in plateau regions—characterized by high intake of ghee, red meat, and low fiber—promote the growth of Collinsella and reduce SCFA - producing bacteria like Ruminococcaceae. These dietary habits may exacerbate gut dysbiosis and inflammation, independent of altitude stress.

[218] Zhao H, Sun L, Liu J, Shi B, Zhang Y, Qu-Zong CR, Dorji T, Wang T, Yuan H, Yang J. Meta-analysis identifying gut microbial biomarkers of Qinghai-Tibet Plateau populations and the functionality of microbiota-derived butyrate in high-altitude adaptation. Gut Microbes. 2024, 16(1), 2350151. doi: 10.1080/19490976.2024.2350151.

Question 4: The description of the SCFA-related molecular route is convoluted and hard to follow; it may need some figures and more precise organization.

Respond 4: Thank you for your valuable input. We have incorporated additional information on the molecular pathways associated with short-chain fatty acids (SCFAs). The detailed modifications are outlined below.

  1. Add text in 3.2.3 Endocrine Pathway:
  • "Epigenetic Switch" Function of SCFAs  

The core pathway of the SCFA-HPA axis involves regulation of the stress response through two primary molecular mechanisms:  

Vagus-Gut-Brain Axis: SCFAs, particularly butyric acid, modulate the stress response by activating intestinal endocrine cell GPR41 receptors and subsequently inhibiting the firing frequency of hypothalamic CRH neurons via vagal afferent fibers [165].  

Epigenetic Regulation: Butyric acid functions as an HDAC inhibitor, enhancing H3K9 acetylation at the glucocorticoid receptor promoter and restoring negative feedback within the HPA axis [138]. In high-altitude environments, SCFA deficiency leads to increased HDAC activity, resulting in methylation of the GR promoter, reduced cortisol sensitivity, and chronic stress.  

  • Plateau-Specific Evidence  

Research using an acute hypoxia model demonstrated that intestinal butyric acid levels in rats were inversely correlated with corticosterone concentrations. Supplementation with butyric acid reversed the hypoxia-induced reduction in GR protein expression [166]. Notably, the Tibetan population exhibits a relatively high abundance of Firmicutes, which may maintain SCFA levels and thereby mitigate excessive HPA axis activation [172].  

Interaction with Neurotransmitters

SCFAs indirectly influence serotonin synthesis by regulating tryptophan metabolism: ① Butyric acid suppresses IDO, reducing the conversion of tryptophan to kynurenine [118].  ② Propionic acid competitively binds to neutral amino acid transporters, enhancing the efficiency of tryptophan transport into the brain.

  1. Add text in1.3 Metabolic Pathways

reduced levels of butyric acid (an important SCFA) weakens gut barrier via downregulating Claudin-1 and inhibits microglial M2 polarization via HDAC6 inactivation [165], leading to increased tryptophan catabolism via IDO activation and HPA axis hyperactivity via GR promoter hypomethylation.

Add Figure 3

Figure 3 The central component of the SCFA molecular pathway.

Note: (1) Impaired Synthesis: The high-altitude environment suppresses the activity of short-chain fatty acid (SCFA)-producing bacteria, such as Firmicutes, resulting in reduced levels of butyric acid and propionic acid; (2) Weakened Barrier Protection: SCFAs maintain the expression of tight junction proteins (e.g., ZO-1, Claudin-1) via the GPR43 receptor. Their deficiency directly contributes to intestinal permeability and barrier dysfunction.  (3) Immune-Neural Interaction Disorder: Butyric acid inhibits the NF-κB signaling pathway, thereby reducing the secretion of pro-inflammatory cytokines IL-6 and TNF-α. Propionic acid activates the GPR43 receptor in microglial cells, promoting M1-type polarization and contributing to neuroinflammation. (4) Uncontrolled HPA Axis: SCFAs regulate the methylation of the CRH promoter through histone deacetylase (HDAC). Their insufficiency impairs cortisol feedback inhibition, leading to dysregulation of the hypothalamic-pituitary-adrenal (HPA) axis.

Question 5: English Language

The work contains a large number of syntactic and grammatical errors. These make things less readable and clear.

Various words must be replaced with appropriate words, i.e., gastroguttract, gastrogutdysfunction, gutmicroorganisms etc

Paragraphs that are too lengthy and have awkward transitions. Divide into manageable chunks for easier reading.

Question 5: Thank you very much for your valuable suggestions. We have carefully reviewed and refined the revised article accordingly.

Round 2

Reviewer 1 Report

Comments and Suggestions for Authors

Dear Authors, thank you for your cooperation in improving your manuscript.

Author Response

We sincerely appreciate your invaluable feedback provided during the review process. 

Reviewer 3 Report

Comments and Suggestions for Authors

The authors have made commendable efforts to address the previously raised concerns; however, significant limitations regarding the quality of the review process and the lack of evidence supporting the proposed theoretical framework remain as below:

A review paper focus on reviewing existing evidence rather than constructing theoretical frameworks without supporting data. Since you agree that the interaction has not been empirically studied and remains speculative, further research needs to be conducted to show the potential interaction of the "plateau environment-gut microbiota-depression" axis (line 828). Therefore, I suggest hypothesizing or presenting evidence-based data rather than making conclusions.

Physiological stress (reference 166) does not equate necessarily psychological stress. These two types of stress are not interchangeable, and there is currently no evidence supporting the proposed axis. While it is noted that plateau environments can cause changes in gut microbiota and that individuals in these environments may have a higher incidence of depression, there is no data to support a direct interaction between these factors. Therefore, I recommend rephrasing the title from “Plateau Environment, Gut Microbiota, and Depression: Unraveling the Concealed Connections” to “Plateau Environment, Gut Microbiota, and Depression: A Possible Concealed Connection?” to better reflect the speculative nature of the content.

Additionally, the methodological approach appears inaccurate as none of the articles in the study include all three keyword combinations as would be expected with an AND selection. Furthermore, none of the studies meet the inclusion criteria outlined in the paper, which require the inclusion of altitudes, microbiota composition or function, and reported depression-related outcomes. 

Author Response

Thank you very much for taking the time to thoroughly review our manuscript. In response to your expert comments, we have addressed each point sequentially in our reply and highlighted the Responseing revisions in blue within the document.

Question 1: A review paper focus on reviewing existing evidence rather than constructing theoretical frameworks without supporting data. Since you agree that the interaction has not been empirically studied and remains speculative, further research needs to be conducted to show the potential interaction of the "plateau environment-gut microbiota-depression" axis (line 828). Therefore, I suggest hypothesizing or presenting evidence-based data rather than making conclusions.

Answer 1: Thank you for your valuable suggestions. We will proceed to implement the following modifications to the article.

  • Minimize excessive speculation in the discussion section.  

Original sentence: "These findings strongly suggest that the high-altitude environment mediates the onset and progression of depression through alterations in the gut microbiota."

The modified: "These findings suggest a potential role for the high-altitude environment in mediating depression through gut microbiota alterations, but direct evidence of this pathway remains limited."

  • Differentiate between physiological stress and psychological stress.  

Original sentence: "This microbial imbalance may be closely linked to the development of acute mountain sickness (AMS), characterized by gut barrier dysfunction [168] and neurotransmitter abnormalities [169], in individuals newly exposed to high altitudes."(line 600)

The modified: "This microbial imbalance may be closely linked to the development of acute mountain sickness (AMS), characterized by gut barrier dysfunction [168]. While AMS involves physiological stress, its direct relationship to psychological stress and depression requires further validation."

  • The abstract should be articulated with precision and restraint.

Original sentence:"In conclusion, while existing evidence supports an association between plateau environments, the gut microbiota and depression, causal mechanisms remain underexplored."

The modified: "In conclusion, existing evidence indicates an association between plateau environments, the gut microbiota, and depression, but causal relationships and underlying mechanisms require further empirical investigation."

  • The discussion section should focus on informed speculation and analysis.  

Original sentence:"The core mechanism by which the high-altitude environment induces depression via the gut microbiota can be summarized as multipathway synergistic dysregulation."

The modified: "A potential core mechanism by which the high-altitude environment may induce depression via the gut microbiota could involve multipathway synergistic dysregulation, though this remains speculative and requires validation."

Question 2: Physiological stress (reference 166) does not equate necessarily psychological stress. These two types of stress are not interchangeable, and there is currently no evidence supporting the proposed axis. While it is noted that plateau environments can cause changes in gut microbiota and that individuals in these environments may have a higher incidence of depression, there is no data to support a direct interaction between these factors. Therefore, I recommend rephrasing the title from “Plateau Environment, Gut Microbiota, and Depression: Unraveling the Concealed Connections” to “Plateau Environment, Gut Microbiota, and Depression: A Possible Concealed Connection?” to better reflect the speculative nature of the content.

Answer 2: Thank you for your valuable suggestion. We will revise the title of the article to "Plateau Environment, Gut Microbiota, and Depression: A Possible Concealed Connection?"

Question 3: Additionally, the methodological approach appears inaccurate as none of the articles in the study include all three keyword combinations as would be expected with an AND selection. Furthermore, none of the studies meet the inclusion criteria outlined in the paper, which require the inclusion of altitudes, microbiota composition or function, and reported depression-related outcomes.

Answer 3: Thank you for your valuable feedback. We have made additional refinements to the method in order to ensure its accuracy. The specific refinements are outlined as follows:  

“We searched PubMed, Web of Science, China National Knowledge Infrastructure (CNKI), and Embase (2000–2025) via the following keyword combinations: ('plateau environment' OR 'high altitude') AND ('gut microbiota' OR 'intestinal flora') AND ('depression' OR 'mental health'). Notably, while this strategy aimed to identify studies addressing all three components, some included studies may focus on associations rather than direct interactions among plateau environment, gut microbiota, and depression. Studies were included if they (1) investigated humans or animal models exposed to altitudes ≥2500 m, (2) measured the gut microbiota composition or function, and (3) reported depression-related outcomes. It is acknowledged that some studies may not explicitly address the interaction between all three factors, and further research is needed to establish direct links.”

Reviewer 4 Report

Comments and Suggestions for Authors

After reading the updated paper, I believe it to be a thorough and organized review that discusses the relationships among depression, gut microbiota, and environmental settings. In accordance with the review report's comments, the writers have included thorough mechanisms, clinical data, and intervention techniques. Nonetheless, there are a few small points where the manuscript's clarity and publishing readiness might be improved with more work. Here are my thoughts and recommendations:

After a few minor edits, the book is ready for publishing and is scientifically sound. The writers ought to:

  1. Take care of the linguistic and clarity problems.
  2. Make sure all terms are used consistently (e.g., "gut microbiota" instead of intestinal flora or gut flora).
  3. Check all citations in figures and tables.

There is no need for significant conceptual or structural adjustments. The manuscript will satisfy the journal's publishing requirements with these improvements.

Comments on the Quality of English Language

Although the work is generally well-written, it will be easier to read with minor grammatical and syntactic corrections. Think about having a professional editing agency or a native English speaker check your work one last time.

Author Response

Reply Reviewer 4

For Reviewer 4:

Thank you very much for taking the time to thoroughly review our manuscript. In response to your expert comments, we have addressed each point sequentially in our reply and highlighted the Responseing revisions in yellow within the document.

Question 1: Take care of the linguistic and clarity problems.

Answer 1: Thank you very much for your suggestions. We have made the following modifications to the language and clarity optimization:

(1) Terminology uniformity and expression standardization

Unify "intestinal flora" as "gut microbiota" to ensure the consistency of terms throughout the text.

Unify "hypobaric hypoxic" as "hypoxic" to avoid confusion of terms.

Unify "lowlands" as "flatlands" to avoid confusion of terms.

  • Sentence structure optimization

A_Original sentence: “Plateau environments pose unique mental health challenges due to stressors such as hypoxia, low temperatures, and intense ultraviolet radiation.”

Optimization “Plateau environments present unique mental health challenges owing to stressors including hypoxia, low temperatures, and intense ultraviolet (UV) radiation.”

B_Original sentence: “These factors drive structural and functional changes in the gut microbiota,  disrupting gut‒brain axis homeostasis and contributing to the increased prevalence of depression in plateau regions compared with lowlands.”

Optimization “These factors induce structural and functional alterations in the gut microbiota,  disrupting gut-brain axis homeostasis and contributing to the higher prevalence of depression in plateau regions relative to flatlands areas.”

Question 2: Make sure all terms are used consistently (e.g., "gut microbiota" instead of intestinal flora or gut flora).

Answer 2: Thank you for your valuable input. We will adopt the term "gut microbiota" throughout the text, instead of using "intestinal flora" or "gut flora."

Question 3: Check all citations in figures and tables.

Answer 3: Thank you for your valuable feedback. We have thoroughly reviewed all citations in the charts and confirmed that they align with the valid entries in the reference list, featuring accurate numbering without any omissions or discrepancies.
